# Correcting misinterpretations of additive models

**Benedict Clark**[1]     **Rick Wilming**[1]     **Hjalmar Schulz**[2]
**Rustam Zhumagambetov**[1]     **Danny Panknin**[1]     **Stefan Haufe**[1,2,3]
[1]Physikalisch-Technische Bundesanstalt, Berlin, Germany
[2]Technische Universität Berlin, Germany
[3]Charité – Universitätsmedizin, Berlin, Germany

## Abstract

Correct model interpretation in high-stakes settings is critical, yet both post-hoc feature attribution methods and so-called intrinsically interpretable models can systematically attribute false-positive importance to non-informative features such as suppressor variables. Specifically, both linear models and their powerful non-linear generalisation such as General Additive Models (GAMs) are susceptible to spurious attributions to suppressors. We present a principled generalisation of activation patterns – originally developed to make linear models interpretable – to additive models, correctly rejecting suppressor effects for non-linear features. This yields PatternGAM, an importance attribution method based on univariate generative surrogate models for the broad family of additive models, and PatternQLR for polynomial models. Empirical evaluations on the XAI-TRIS benchmark with a novel false-negative invariant formulation of the earth mover's distance accuracy metric demonstrates significant improvements over popular feature attribution methods and the traditional interpretation of additive models. Finally, real-world case studies on the COMPAS and MIMIC-IV datasets provide new insights into the role of specific features by disentangling genuine target-related information from suppression effects that would mislead conventional GAM interpretations.

## 1   Introduction

As machine learning (ML) models are increasingly used in high-stakes domains, the need for reliable model explanations has grown in tandem, with much of the work in explainable AI (XAI) has focused on interpreting ML models through post-hoc attribution techniques. Parallel to this, simple model architectures such as Generalised Additive Models (Hastie & Tibshirani, 1986; Rudin, 2019), which are thought of as 'inherently interpretable' have become popular. GAMs non-linearly model the output as a sum of smooth functions applied to each input feature, followed by a link function – typically the logit or softmax in classification settings. This architecture is often considered interpretable in the sense that each component function – also known as a shape function – can be visualised to illustrate how the feature contributes to the model's prediction. However, how models use specific features, and therefore, how model coefficients or shape functions must be interpreted, depends non-trivially on the data generating process, even when models are decomposable into simple terms. It is often erroneously assumed that non-zero coefficients corresponding to a feature or feature pair imply that these features carry information about the prediction target. However, depending on the joint causal structure of the model's in- and outputs, optimal models may assign significant non-zero weight to non-informative features and, conversely, zero weight to informative features (Haufe et al., 2014). It is also possible that a feature enters the model with a polarity that opposes the correlation between the feature and the target. Such effects can be due to suppressor variables, which are variables that are statistically unrelated or weakly related to the prediction target but help the model to remove variance from other variables, therefore improving its predictions (see, e.g., Conger, 1974; Weichwald et al., 2015; Haufe et al., 2024).

39th Conference on Neural Information Processing Systems (NeurIPS 2025).

In the context of linear models, Haufe et al. (2014) highlighted that model weights cannot be interpreted as typically desired unless adjusted by the data covariance, which yields the linear activation pattern, or simply *Pattern*. Without such adjustment, suppressor variables can receive arbitrarily large weights of any polarity, misleadingly signalling importance. Such misleading attribution is also systematically observed in a large variety of popular feature attribution methods applied to Bayes-optimal model in the presence of suppressors Wilming et al. (2023). Empirically, Wilming et al. (2022) showed that the Pattern approach outperforms other XAI methods in linear problem settings in the presence of suppressors, while Clark et al. (2024b) extended this analysis to non-linear problems. Addressing the shortcomings of existing work on interpretable non-linear models, our contributions are as follows:

1. We introduce a framework for computing univariate and bivariate feature importances to turn the broad class of non-linear GAMs into models that can be interpreted in terms of practically relevant associations between features and prediction targets.

2. We instantiate this framework for multiple arts of models, resulting in two new explanation approaches: PatternQLR (for quadratic models) and PatternGAM (for general additive models).

3. We empirically validate our methods on the XAI-TRIS benchmark suite, reporting superior performance of PatternGAM and PatternQLR over GAMs and existing attribution methods.

4. We present a novel metric for assessing the performance of feature attributions, false-negative invariant earth mover's distance accuracy (FNI-EMDA), to overcome the tendency of vanilla EMD metrics to penalise false-negative attributions and operate over a narrow resolution.

5. We apply our approaches to two real-world datasets for the prediction of recidivism risk and in-hospital mortality. Using PatternGAM, we unearth new insights into the role of sensitive attributes in the prediction process, thereby revealing critical misinterpretations of conventional GAM-based interpretations that can be attributed to suppression.

## 2    Background: additive models and suppressor variables

Generalised linear models (GLM) $g(E[y|\mathbf{x}]) = \mathbf{w}^\top \mathbf{x} + \epsilon$, where $\mathbf{x} \in \mathbb{R}^D$ are input features, are often seen as inherently interpretable due to their simple structure, assigning one weight $w_i$ to each feature $x_i$. Consider the two-dimensional linear structural model introduced by Wilming et al. (2023):

$$\mathbf{x} = \mathbf{a}y + \boldsymbol{\eta} \,, \tag{1}$$

where $y$ is the target label, $\mathbf{a} = (1,0)^\top$ is the signal activation pattern defining how the signal is represented in each feature, and $\boldsymbol{\eta} \sim \mathcal{N}(0, \boldsymbol{\Sigma}_\eta)$. The data covariance is defined as $\boldsymbol{\Sigma}_\mathbf{x} = \begin{bmatrix} r_1^2 + 1 & cr_1 r_2 \\ cr_1 r_2 & r_2^2 \, , \end{bmatrix}$ for non-negative standard deviations $r_1$ and $r_2$. Then, with $\mu_1 = (1,0)^\top$ and $\mu_2 = (-1,0)^\top$, the weights of the Bayes-optimal GLM are calculated as $\mathbf{w} = \boldsymbol{\Sigma}_\eta^{-1}(\mu_1 - \mu_2)$, which derives to $\mathbf{w} = (\alpha, -\alpha cr_1/r_2)^\top$ for $\alpha := (1 + (cr_1/r_2)^2)^{-1/2}$, showing that, while the suppressor feature $x_2$ contains no class-discriminative information about $y$, it must still receive non-zero weight when the correlation between features is non-zero. This misleading attribution to suppressors is corrected by the *activation pattern* or *Pattern* (Haufe et al., 2014), defined as $\hat{\mathbf{a}} = \boldsymbol{\Sigma}_\mathbf{x} \mathbf{w} \left( \mathbf{w}^\top \boldsymbol{\Sigma}_\mathbf{x} \mathbf{w} \right)^{-1}$, which is the ordinary least-squares (OLS) estimate of the coefficients $a_i$ of the univariate models $a_i \hat{y} + c_i = x_i$ and simplifies to the univariate solutions $a_i = \text{Cov}[x_i, \hat{y}] \text{Var}[\hat{y}]^{-1}$. In the present case, $\hat{\mathbf{a}} = (1,0)^\top$, correctly reflecting the lack of a statistical association between $x_2$ and $y$ and forming the desired data-aware global explanation.

**Generalised Additive Models**   Generalised additive models (GAMs, Hastie & Tibshirani, 1986; Lou et al., 2013; Nori et al., 2019) model the prediction target $y \in \mathbb{R}$ as

$$g(E[y|\mathbf{x}]) = \sum_i^D f_i^{\text{GAM}}(x_i) + \sum_{j<k}^D f_{jk}^{\text{GAM}}(x_j, x_k) \,, \tag{2}$$

where $f_i^{\text{GAM}}$ and $f_{jk}^{\text{GAM}}$ are univariate and bivariate *shape functions* applied to single features and feature pairs, respectively, and where $g$ is a link function, often chosen as the logit function in binary

classification tasks and the identity function in regression tasks. In the following, we focus on binary classification, although all presented results should hold for other suitable choices of link functions.

In the original GAM and GA$^2$Ms approaches (Hastie & Tibshirani, 1986; Lou et al., 2013), each shape function is represented as a linear combination of basis functions such as splines. Explainable Boosting Machines (EBMs) replace the spline functions with tree-based bagging and boosting techniques for particularly expressive and 'wiggly' functions, whereas Neural Additive Models (NAMs) (Agarwal et al., 2021) replace the spline- or tree-based shape functions with small neural networks. Model parameters are fitted using training data $\mathcal{D} = \{(\mathbf{x}^{(1)}, y^{(1)}) \ldots, (\mathbf{x}^{(N)}, y^{(N)})\}$, enabling coordination among the individual functions during optimisation. In the following, we assume that $E[\mathbf{x}] = \mathbf{0}$ and $\forall i \, \mathrm{Var}[x_i] > 0$. Expectations and variances are taken across training samples. We refer to single entries $x_j$ and pairs $(x_j, x_k)$ of $\mathbf{x}$ as univariate and bivariate input features, respectively, and to $f_j$ and $f_{j,k}$ as univariate and bivariate functions. Note that we largely refrain from referring to these functions as modelling interactions per se, as the actual computation carried out depends on statistical properties of the input features $x_j$ and $x_k$.

We here choose NAMs as our main additive model architecture, where we extend the original approach to also include bivariate terms $f_{j,k}^{\mathrm{GAM}}$. Practically this amounts to adding an additional two-input sub-network with one output per modelled feature combination. In practice, rather than choosing the entire quadratic set of all possible bivariate feature combinations, Lou et al. (2013) propose and Nori et al. (2019) make use of the FAST algorithm, which selects bivariate terms based on the residual of the best current additive model in feature space. Feature pairs are iteratively added to the model until there is no gain in accuracy. This approach is adopted here as well.

Fitted shape functions $f_i^{\mathrm{GAM}}$ are often visualised to show how the model output varies as the value of a single feature or feature pair changes. Non-zero shape functions are interpreted as indicating a statistical association between feature and target, and their scales and signs are interpreted as strengths and directions of the associations (Lou et al., 2012; Caruana et al., 2015, e.g.,). Following Haufe et al. (2014), these interpretations are theoretically unjustified and potentially misleading. This is because GAM shape functions reflect how individual features need to be transformed to yield the desired target – but are not based on statistical association with the target. In the two-dimensional suppressor setup introduced above, fitting a GAM with the identity link function $g$ leads to a non-zero linear function in $f_2^{\mathrm{GAM}}(x_2)$. Even though $x_2$ is not discriminative in isolation, its statistical association with $x_1$ (the true signal carrier) through the shared noise component causes the GAM to model a non-zero functional effect for $x_2$, resulting in a spurious attribution of importance.

**Existing Non-linear Pattern Approaches**   Thus far, no analogy to the linear activation pattern has been proposed for GAMs, while proposals to make other families of non-linear models interpretable present disadvantages. PatternNet and PatternAttribution are two well known approaches which apply the principles of Pattern to non-linear models, aiming to trace signal flow in deep neural networks (Kindermans et al., 2018). While reducing to Pattern in linear cases, both have been shown to have degraded performance on controlled non-linear benchmarks, particularly when correlated background noise acts as a suppressor (Clark et al., 2024b). The Kernel Pattern method of Zhang et al. (2024) computes the Pattern in kernel space, but due to the pre-image problem (Bakir et al., 2003; Honeine & Richard, 2009), the corresponding input-space Pattern is not uniquely defined and must be estimated. This reverse mapping is often ill-posed, unstable, or out-of-distribution, and the method captures only implicit kernel-induced relationships (e.g., similarity) rather than explicit or flexible feature combinations like those learned by additive models with explicit bivariate terms. Finally, recent work by Gjølbye et al. (2025) uses the Pattern concept in the context of locally linear explanations.

## 3   Methodology

In the following, we extend the Pattern approach to make two families of additive non-linear models, quadratic logistic regression (QLR) and GAMs, interpretable with respect to the Statistical Association Property (Wilming et al., 2022, 2024).

### 3.1   Pattern Quadratic Logistic Regression (PatternQLR)

We first extend the idea of activation patterns to a polynomial expansion of the input $\mathbf{x}$ up to degree 2, consisting of all first-order terms and all unique pairwise products

$$\mathbf{z}^{\text{QLR}} = \phi(\mathbf{x}) = \left(x_1, \ldots, x_D, x_1^2, \ldots, x_D^2, x_1 x_2, \ldots, x_{D-1} x_D\right)^\top . \tag{3}$$

Note that we include only one of the terms $x_j x_k$ and $x_k x_j$ for each feature pair (i.e., we assume a fixed ordering such that $j \leq k$), avoiding redundant terms due to symmetry. This results in $D^{\text{QLR}} = 2D + (D^2 - D)/2$ ($D$ linear, $D$ quadratic and $(D^2 - D)/2$ pairwise) overall features that capture all uni- and bivariate effects. For simplicity, we define $f_j^{\text{QLR}}(x_j) = x_j$, $f_{j,k}^{\text{QLR}}(x_j, x_k) = x_j x_k$, and $f_i^{\text{QLR}}(\mathbf{x}) = z_i^{\text{QLR}}$. This Quadratic Logistic Regression (QLR) mapping approach (if we were to use the full non-redundant set of pairwise products) is equivalent to a degree-2 polynomial kernel $k(\mathbf{x}_i, \mathbf{x}_j) = (\gamma \mathbf{x}_i^\top \mathbf{x}_j + c)^2$ with $\gamma = 1, c = 0$. Both the QLR and this particular polynomial kernel can be viewed as additive models with multiplicative feature combinations.

To solve the classification problem, we train a linear classifier (e.g., linear logistic regression, LLR) on $\mathbf{z}^{\text{QLR}}$. From the obtained weight vector $\mathbf{w}^{\text{QLR}}$ in $\mathbf{z}$-space, we can compute the QLR activation pattern (PatternQLR) $\mathbf{a}^{\text{PQLR}} = \text{Cov}[\mathbf{z}^{\text{QLR}}, g(\hat{y}^{\text{QLR}})]\text{Var}[g(\hat{y}^{\text{QLR}})]^{-1}$, where $g(\hat{y}^{\text{QLR}}) = \mathbf{w}^{\text{QLR}\top}\mathbf{z}^{\text{QLR}} + u^{\text{QLR}}$. This is again equivalent to fitting univariate models $a_i^{\text{PQLR}} g(\hat{y}) + v_i^{\text{PQLR}} = z_i^{\text{QLR}}$ post-hoc with OLS.

In cases where all features of the extended feature vector $\mathbf{z}^{\text{QLR}}$ possess a similar scale, the scale of the $a_i^{\text{PQLR}}$ is also comparable across features and can serve as a quantitative measure of feature importance. This is the case in the neuroimaging context, in which Pattern has been introduced, where features typically correspond to sensors of the same type and are measured in the same units (Haufe et al., 2014). If features are on different scales (e.g. correspond to different physical quantities), the coefficients $a_i^{\text{PQLR}}$ are not quantitatively comparable across features as well as across linear to quadratic terms. Here we address this issue in two ways. First, we standardise each $z_i^{\text{QLR}}$ to zero mean and unit variance based on the training data prior to Pattern computation. Second, we derive additional feature importance metrics that are scale-invariant. To this end, we fit $D^{\text{QLR}}$ univariate LLR models of the form $g(E[y|\mathbf{x}]) = f_i^{\text{PQLR}}(\mathbf{x}) = b_i^{\text{PQLR}} z_i^{\text{QLR}} + d_i^{\text{PQLR}}$, where the univariate *PatternQLR shape functions* $f_i^{\text{PQLR}}(\mathbf{x})$ evaluate to $f_i^{\text{PQLR}}(\mathbf{x}) = b_i^{\text{PQLR}} x_j + d_i^{\text{PQLR}}$ for univariate features and to $f_i^{\text{PQLR}}(\mathbf{x}) = b_i^{\text{PQLR}} x_j x_k + d_i^{\text{PQLR}}$ for bivariate features. Rather than interpreting scale-dependent coefficients $b_i^{\text{PQLR}}$ directly, we here interpret shape functions $f_i^{\text{PQLR}}$, which can be visualised as functions of the underlying input-space features $x_j$ and $x_k$. Note that PatternQLR shape functions are fit to the true class labels $y$ instead of their QLR estimates $\hat{y}^{\text{QLR}}$ as done in the Pattern approach.

### 3.2 PatternGAM

For the fitted additive model Eq. (2), we obtain $D^{\text{GAM}} = D + (D^2 - D)/2$ ($D$ univariate and $(D^2 - D)/2$ bivariate) non-linear features produced by shape functions $f_i^{\text{GAM}}(\mathbf{x})$ that are jointly learned. We collect these in

$$\mathbf{z}^{\text{GAM}} = \phi^{\text{G}}(\mathbf{x}) = \left(f_1^{\text{G}}(x_1), \ldots, f_D^{\text{G}}(x_D), f_{1,2}^{\text{G}}(x_1, x_2), \ldots, f_{D-1,D}^{\text{G}}(x_{D-1}, x_D)\right)^\top , \tag{4}$$

where, in practice, $\mathbf{z}^{\text{GAM}}$ is estimated as the mean over $K = 100$ NAM fits with random initialisations, and a subset of bivariate terms are selected by FAST. Summing up the entries of $\mathbf{z}^{\text{GAM}}$ according to Eq. (2) formally amounts to applying a linear logistic regression model with fixed weights $\mathbf{w}^{\text{GAM}} \equiv \mathbf{1}$. We observe, however, that this results in slightly suboptimal models, presumably due to the model averaging process. Therefore, we re-estimate $\mathbf{w}^{\text{GAM}}$ using LLR leading to adjusted features $z_i^{\text{GAM}} \leftarrow w_i^{\text{GAM}} z_i^{\text{GAM}}$, which are used in all subsequent steps. After a further standardisation of the $z_i^{\text{GAM}}$, we again calculate the PatternGAM activation pattern vector as $\mathbf{a}^{\text{PGAM}} = \text{Cov}[\mathbf{z}^{\text{GAM}}, g(\hat{y}^{\text{GAM}})]\text{Var}[g(\hat{y}^{\text{GAM}})]^{-1}$. Similar to PatternQLR, we also fit separate univariate LLR models for each of the $D^{\text{GAM}}$ features, where $f_i^{\text{PGAM}}(x_j) = b_i^{\text{PGAM}} f_j^{\text{GAM}}(x_j) + d_i^{\text{PGAM}}$ for univariate features and $f_i^{\text{PGAM}}(x_j, x_k) = b_i^{\text{PGAM}} f_{j,k}^{\text{GAM}}(x_j, x_k) + d_i^{\text{PGAM}}$ for bivariate features.

### 3.3 Feature importance metrics

Our main object of feature interpretation are PatternQLR and PatternGAM shape functions $f_i^{\text{PQLR/PGAM}}$, evaluated as functions of uni- or bivariate features, which we compare to their QLR/GAM counterparts $f_i^{\text{QLR/GAM}}$. In addition, we use the following scalar metrics of feature importance.

**Activations patterns:** We use the absolute value of the PatternQLR/PatternGAM coefficients

$a_i^{\text{PQLR/PGAM}}$ as estimated from standardised features $z_i^{\text{QLR/GAM}}$: $\text{PAT}_i(f_i^{\text{QLR/GAM}}) = |a_i^{\text{PQLR/PGAM}}|$.

**Scale:** We evaluate the standard deviation of shape function values as $\text{SD}_i(f_i) = \text{Var}[f_i(\mathbf{x})]^{1/2}$ across training samples for both the original QLR and GAM shape functions $f_i^{\text{QLR/GAM}}$ and their PatternQLR and PatternGAM counterparts $f_i^{\text{PQLR/PGAM}}$.

**Discriminability:** We use the area under the receiver operating curve (AUROC) to assess the univariate discriminability of each univariate or bivariate shape function. We define $\text{DISCR}_i(f_i, y) = 2(\text{AUROC}(f_i, y) - 0.5)$, where positive associations with the target are reflected by scores between 0 and 1. Being a correlation measure, DISCR is invariant to rescaling of $f_i$, and presents a general way to avoid suppressor misattribution in additive models.

**Intersection:** Our main interpretation goal is to identify features or feature pairs that are both informative w.r.t. the target as well as 'used' by the model. The former can be ensured by measuring importance either with the PAT or DISCR metrics, or by evaluating SD for $f^{\text{PQLR/PGAM}}$. As QLR and GAM models have no incentive to estimate non-zero shape functions for unnecessary, e.g. redundant, features, the latter is also ensured in principle. In practice, however, such features may receive small non-zero weights, which could preserve any potential statistical association with the target. Such features will receive strong importance according to all the above metrics, despite not being used by the model. To counteract this behavior, we also introduce $\text{PROD}_i = \text{SD}_i(f_i^{\text{QLR/GAM}}) \cdot \text{SD}_i(f_i^{\text{PQLR/PGAM}})$.

Finally, while all importance metrics are evaluated on both univariate and bivariate features, we restrict our statistical evaluation to input-space. To this end, we aggregate univariate and bivariate feature importances. We define $\text{IMP}_j = \text{IMP}_i(f_i)$ for $z_i = f_j(x_j)$ and $\text{IMP}_{j,k} = \text{IMP}_i(f_i)$ for $z_i = f_i(x_j, x_k)$, where $\text{IMP} = \{\text{PAT}, \text{SD}, \text{DISCR}\}$ (analogously for INTER). The aggregated input-space importance $\text{IMP\_AGG} \in \mathbb{R}^D$ is denoted by $\text{IMP\_AGG}_j = \max_{k \neq j}(\text{IMP\_PAIR}_{j,k})$, where

$$\text{IMP\_PAIR}_{j,k} = \begin{cases} \text{IMP}_j & \text{IMP}_j < \text{IMP}_k \\ \max(\text{IMP}_{j,k}, \text{IMP}_j) & \text{else}, \end{cases} \qquad (5)$$

with $f_{j,k} := f_{k,j}$ for the case $j > k$, and where $\text{IMP} = \{\text{PAT}, \text{SD}, \text{DISCR}, \text{INTER}\}$. We compare these metrics with existing feature attribution methods acting as baselines (c.f. Section 4.1). For local methods, we define the global attribution as the average absolute local attribution over all samples. Finally, denoting by $\mathbf{s} \in \mathbb{R}^D$ a global attribution, we rectify all entries by taking their absolute values, followed by a division by the sum of all entries, thereby ensuring that $\forall i \, s_i \geq 0$ and $\sum_i s_i = 1$.

## 3.4 Theoretical Properties PatternQLR and PatternGAM

Let $s(\mathcal{D})$ be a method capable of generating global input-space attributions $\mathbf{s} \in \mathbb{R}^D$.

**Definition 1 (Statistical Association Property, SAP, Wilming et al. (2022, 2024))** *A feature attribution method $s$ possesses the SAP if any significant non-zero importance attribution to a univariate feature $x_j$ indicates a statistical association with the target: $s_j$ indicates importance $\Rightarrow x_j \not\perp y$.*

The SAP rules out that non-informative variables, including suppressors, are assigned significant importance, which is a prerequisite for correct interpretations and use of attributions in downstream tasks such as model debugging, scientific discovery, and counterfactual analysis (Haufe et al., 2024).

**Theorem 1** *The following quantities possess the SAP.*

1. *PatternQLR/GAM shape function coefficients $b_i^{\text{PQLR/PGAM}}$ for (possibly) non-linear univariate features $z_i^{\text{QLR/GAM}} = f_j^{\text{QLR/GAM}}(x_j)$.*

2. *Under conditions, PatternQLR/GAM coefficients $a_i^{\text{PQLR/PGAM}}$ for $z_i^{\text{QLR/GAM}} = f_j^{\text{QLR/GAM}}(x_j)$.*

3. *PatternQLR/GAM shape functions $f_j^{\text{PQLR/PGAM}}(x_j)$ and their standard deviations $SD(f_j^{\text{PQLR/PGAM}})$.*

4. *Discriminability metrics applied to univariate functions $f_j(x_j)$, e.g., $f_j^{\text{GAM}}$.*

**Theorem 2** *The original GAM shape functions $f_i^{\text{GAM}}$ for $z_i = x_j$ and their standard deviations $SD(f_i^{\text{GAM}})$ do not possess the SAP.*

The proofs are presented in Appendix A.

## 3.5 Explanation Performance Metrics

We conduct ground-truth benchmarking of univariate and bivariate feature attributions using the XAI-TRIS benchmarking suite (Clark et al., 2024b, see also Supplementary Section D). In the LIN

and MULT XAI-TRIS scenarios, the features $x_j$ for which the SAP holds are known by construction and are collected in the set $\mathcal{A}^+$, where $j \in \mathcal{A}^+ \Leftrightarrow x_j \not\perp\!\!\!\perp y$. Analogously, we define the ground-truth set of pairwise interactions $\mathcal{I}^+$ for the XOR scenario. In this work, we aggregate univariate and bivariate importance attributions in input space for benchmarking purposes. In this setting, the ground-truth is given by the set $\mathcal{T}^+ = \{j \mid j \in \mathcal{A}^+ \ \lor \ \exists k \ (j,k) \in \mathcal{I}^+\}$ containing all univariate input features that are either in $\mathcal{A}^+$ or part of a feature pair that is in $\mathcal{I}^+$. We further define the binary vector $\mathbf{t}^+ \in \mathbb{R}^D$, where $t_j^+ = {}^1\!/\!|\mathcal{T}^+| \Leftrightarrow x_j \in \mathcal{T}^+$ and $t_j^+ = 0$ otherwise. Based on the known ground truth, the *explanation performance* of a given explanation $\mathbf{s}$ is assessed using the following metrics.

**Importance Mass Accuracy (IMA):** Defined as $\mathrm{IMA}(\mathbf{s}, \mathcal{T}^+) = \left(\sum_{t \in \mathcal{T}^+} s_t\right) / \left(\sum_{d=1}^{D} s_d\right)$, this is the proportion of explanation 'mass' assigned to known ground-truth features compared to the total importance attributed to all features (Arras et al., 2022; Clark et al., 2024b).

**False-Negative-Invariant Earth Mover's Distance Accuracy (FNI-EMDA):** For image data, the Euclidean distance between input pixels can be taken into account when evaluating explanation performance (Clark et al., 2024b). Given such a distance $\mathbf{C} \in \mathbb{R}^{D \times D}$, the earth-mover's distance (EMD) between the continuous-valued attribution $\mathbf{s}$ and the ground-truth importance $\mathbf{t}^+$ is defined as

$$\mathrm{EMD}(\mathbf{s}, \mathbf{t}^+, \mathbf{C}) = \min_{\boldsymbol{\Gamma} \in \mathbb{R}^{D \times D}} \sum_{u=1}^{D} \sum_{v=1}^{D} \Gamma_{u,v} C_{u,v} \quad \text{s.t.} \quad \boldsymbol{\Gamma} \mathbf{1} = \mathbf{s}, \boldsymbol{\Gamma}^\top \mathbf{1} = \mathbf{t}^+ , \tag{6}$$

where $\boldsymbol{\Gamma} \in \mathbb{R}^{D \times D}$ is a transport plan. Unlike IMA, the EMD can readily distinguish between minor local displacements of ground-truth importance and globally altered importance patterns. EMD, however, requires that the explanation is transformed into the complete ground truth; that is, to avoid penalisation, XAI methods need to assign equally high importance to every ground-truth input feature. In practice, subsets of important features may form equally valid explanations – just as doctors, models may not need to use every symptom or available measurement to diagnose a patient. False-Negative-Invariant EMD Accuracy (FNI-EMDA) addresses this shortcoming by setting the transport costs between all ground-truth features to zero, which is achieved via the modified Euclidean distance $C'_{u,v} = C_{u,v} \left(1 - \mathbb{I}(u \in \mathcal{T}^+ \land v \in \mathcal{T}^+)\right)$. FNI-EMDA is then defined as the negative normalised EMD with modified ground distance $\mathbf{C}'$: $\mathrm{FNI\text{-}EMDA}(\mathbf{s}, \mathbf{t}^+, \mathbf{C}') = 1 - {}^{\mathrm{EMD}(\mathbf{s}, \mathbf{t}^+, \mathbf{C}')}\!/\!_{\delta_{\max}}$, where $\delta_{\max} = \max_i[(\mathbf{C}' \mathbf{t}^+)_i]$ is the worst-case EMD for a given ground-truth $\mathbf{t}^+$. This novel metric does not unfairly penalise subset-style attributions, as illustrated in Appendix Section C, and improves on the original formulation of EMD accuracy by providing a higher resolution of possible resulting scores. We analogously define EMD Accuracy (EMDA) by replacing $\mathbf{C}'$ again with the unaltered Euclidean distance $\mathbf{C}$.

## 4 Results

Code for all experimental results is provided[1] with details on usage, data availability, and compute required available in Appendix Section B. We acknowledge the use of LLMs (ChatGPT 4o and Gemini 2.5 pro) for early prototyping, plot structuring and formatting, however all uses have been verified and tested (and compared with known results where possible).

### 4.1 Experiments on XAI-TRIS synthetic ground truth imaging data

Agarwal et al. (2021) motivate the use of NAMs for computer vision tasks due to their potential intelligibility. The XAI-TRIS (Clark et al., 2024b) benchmark datasets provide visually compelling binary classification scenarios with known signal patterns, shown as 'tetrominos' (Golomb, 1996). Here, tetromino signals $\mathbf{a}$ are combined additively or multiplicatively with two different types of noisy backgrounds $\boldsymbol{\eta}$: white noise, or correlated noise smoothed with a Gaussian filter. Different binary classification scenarios are presented – a linear additive noise task (LIN) , a multiplicative noise task (MULT), and an XOR task with additive noise. In all cases, signal patterns $\mathbf{a}$ are combined with a background noise component $\boldsymbol{\eta}$, where the background is either white (Gaussian) noise, or correlated noise (white noise smoothed by a 2D Gaussian filter). The correlated background noise induces suppressors due to the spatial overlap of the background noise and the tetromino components. Here, we extend this with an explicit distractor pattern $\mathbf{d}$ that spatially overlaps the signal tetrominos,

---

[1]`https://github.com/braindatalab/pattern-gam`

again acting as a suppressor. The full details and visualisations of data generation, model parameters, and experimental setup can be found in Appendix D.

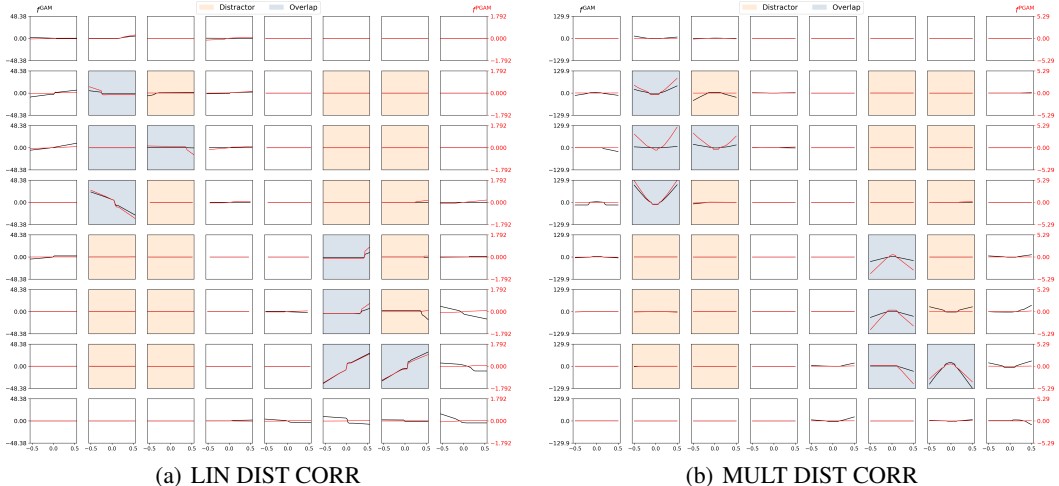

(a) LIN DIST CORR

(b) MULT DIST CORR

Figure 1: GAM vs PatternGAM shape functions for a NAM trained on the LIN (left) and MULT (right) XAI-TRIS scenarios with dedicated distractor patterns and the correlated (CORR) noise background. Grey shaded features represent class-informative ground truth tetrominos, which completely overlap with the non-informative distractor (suppressor) patterns (in orange). While the raw shape functions, used as the traditional global explanation of additive models, highlight noise features and distractors, PatternGAM correctly removes their influence on the explanation.

In the LIN and MULT scenarios, no interactions are present by construction, for which reason we do not include explicit bivariate feature terms in the underlying model. Figure 1 shows the raw shape functions for a NAM trained over these scenarios, where a dedicated distractor (suppressor) pattern, shaded in orange, overlaps with the tetromino signals in grey. One can see that, when PatternGAM is applied, the shape functions of irrelevant features are nullified, overcoming misleading attribution of importance to distractor features as seen in the original GAM shape functions.

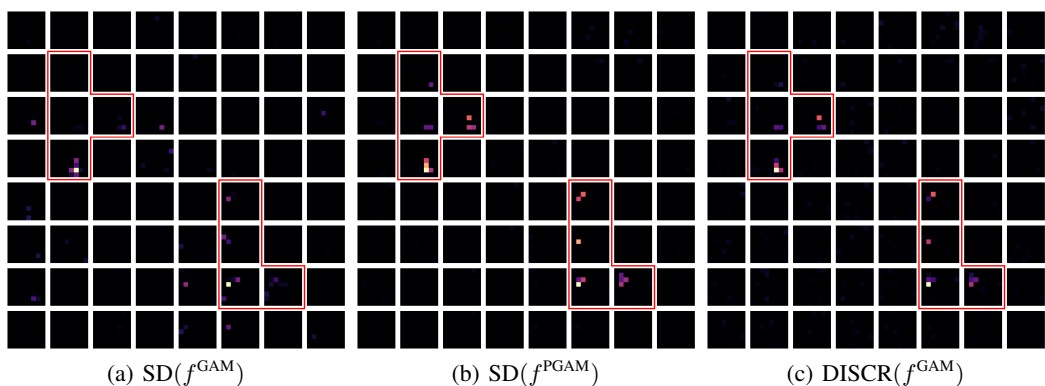

(a) $\text{SD}(f^{\text{GAM}})$      (b) $\text{SD}(f^{\text{PGAM}})$      (c) $\text{DISCR}(f^{\text{GAM}})$

Figure 2: Spatial maps of bivariate feature importances for 256 feature pairs in the XOR-DIST-CORR XAI-TRIS scenario. Heat indicates higher importance and red contours mark ground-truth features. Importance of the original NAM shape functions in (a), measured by the SD metric, shows diffuse and noisy characteristics, whereas PatternGAM importance (b) concentrates within the ground-truth regions, showing the cross-tetromino interaction pattern of the underlying XOR problem. Similarly, the discriminability map (c) highlights ground-truth feature pairs as the most informative.

Figure 2 shows spatial importance maps for bivariate feature combinations in terms of the standard deviation $\text{SD}(f^{\text{GAM}})$ and $\text{SD}(f^{\text{PGAM}})$ of the NAM and PatternGAM shape functions, as well as the univariate discriminability $\text{DISCR}(f^{\text{GAM}})$. This is shown for the XOR-DIST-CORR XAI-TRIS

setting trained with 128 bivariate terms as selected using FAST. The 'picture-in-picture' style image shows how each feature of the $8\times8$ input image is estimated to interact with all other features. For SD($f^{GAM}$), limited structure is visible; however for SD($f^{PGAM}$) and DISCR($f^{GAM}$) we see the ground-truth XOR interaction between the T-shaped tetromino in the top left and the L-shaped tetromino in the bottom right reflected in the importances attributed to the corresponding pixel pairs.

We further qualitatively and quantitatively evaluate the global explanations of PatternGAM and PatternQLR with those of the original NAM, an EBM, Kernel Pattern, SHAP (Lundberg & Lee, 2017), Integrated Gradients (Sundararajan et al., 2017), PatternNet and PatternAttribution. The latter four are applied to a four-layer Multi-layer Perceptron (MLP). Appendix Figure 8 shows that PatternGAM and PatternQLR tend to highlight truly important features more consistently than all other approaches. Interestingly, PatternNet and Kernel Pattern highlight smoothed correlated noise in several CORR scenarios, and all methods that do not possess the SAP can highlight significant attribution to the distractor features.

Tables 1 presents results of a quantitative assessment of the explanation performance of all attributions on the XAI-TRIS data using the FNI-EMDA metric. Analogous results for the IMA and regular EMDA metrics are shown in Appendix Tables 3 and 4. Generally, the standard interpretations from the NAM and EBM as well as the traditional post-hoc feature attribution methods SHAP and Integrated Gradients have worse performance than Pattern-based methods, showing false-positive attribution to suppressor and otherwise uninformative features.

| Method | LIN | | | | MULT | | | | XOR | | | |
|---|---|---|---|---|---|---|---|---|---|---|---|---|
| | WHITE | CORR | DIST WHITE | DIST CORR | WHITE | CORR | DIST WHITE | DIST CORR | WHITE | CORR | DIST WHITE | DIST CORR |
| PAT($f^{GAM}$)† | 0.95±0.01 | 0.80±0.04 | 0.96±0.01 | 0.80±0.04 | 0.94±0.01 | 0.80±0.04 | 0.94±0.01 | 0.81±0.04 | 0.93±0.01 | 0.84±0.04 | 0.92±0.01 | 0.84±0.05 |
| PAT($f^{QLR}$)† | 0.88±0.00 | 0.83±0.02 | 0.91±0.00 | 0.78±0.01 | 0.89±0.00 | 0.73±0.00 | 0.89±0.00 | 0.72±0.00 | 0.92±0.00 | 0.89±0.01 | 0.91±0.00 | 0.88±0.03 |
| SD($f^{PGAM}$)† | 0.99±0.01 | 0.96±0.02 | **1.00±0.00** | 0.94±0.02 | 0.98±0.01 | 0.96±0.02 | 0.98±0.01 | 0.96±0.02 | 0.97±0.01 | 0.97±0.01 | 0.96±0.01 | 0.97±0.01 |
| SD($f^{PQLR}$)† | 0.92±0.00 | 0.93±0.03 | 0.95±0.00 | 0.86±0.02 | 0.96±0.00 | 0.76±0.00 | 0.96±0.00 | 0.76±0.00 | 0.96±0.00 | 0.98±0.01 | 0.95±0.00 | 0.97±0.01 |
| DISCR($f^{GAM}$)† | 0.97±0.01 | 0.93±0.02 | 0.97±0.01 | 0.90±0.02 | 0.91±0.01 | 0.88±0.03 | 0.91±0.01 | 0.88±0.03 | 0.96±0.00 | 0.96±0.01 | 0.94±0.01 | 0.94±0.02 |
| DISCR($f^{QLR}$)† | 0.90±0.00 | 0.82±0.01 | 0.91±0.00 | 0.81±0.01 | 0.90±0.00 | 0.76±0.00 | 0.90±0.00 | 0.76±0.00 | 0.93±0.00 | 0.94±0.01 | 0.91±0.00 | 0.93±0.01 |
| PROD($f^{P/GAM}$)† | **1.00±0.00** | **0.99±0.01** | **1.00±0.00** | **0.99±0.01** | **1.00±0.01** | **0.99±0.01** | **1.00±0.00** | **0.99±0.01** | **1.00±0.00** | 0.99±0.01 | **1.00±0.00** | **1.00±0.00** |
| PROD($f^{P/QLR}$)† | 0.99±0.00 | **0.99±0.01** | **1.00±0.00** | **0.99±0.00** | **1.00±0.00** | 0.82±0.00 | **1.00±0.00** | 0.83±0.00 | **1.00±0.00** | **1.00±0.00** | **1.00±0.00** | **1.00±0.00** |
| SD($f^{GAM}$) | 0.98±0.01 | 0.89±0.02 | 0.96±0.01 | 0.88±0.02 | 0.97±0.01 | 0.89±0.02 | 0.97±0.02 | 0.89±0.01 | 0.88±0.02 | 0.85±0.04 | 0.90±0.02 | 0.87±0.03 |
| SD($f^{QLR}$) | 0.85±0.00 | 0.87±0.00 | 0.87±0.01 | 0.89±0.00 | 0.90±0.00 | 0.75±0.00 | 0.90±0.00 | 0.75±0.00 | 0.87±0.01 | 0.90±0.00 | 0.85±0.01 | 0.91±0.00 |
| EBM | 0.96±0.00 | 0.90±0.00 | 0.92±0.00 | 0.91±0.00 | 0.96±0.00 | 0.90±0.00 | 0.96±0.00 | 0.89±0.00 | 0.66±0.03 | 0.59±0.04 | 0.68±0.04 | 0.62±0.08 |
| Kernel Pattern | 0.96±0.00 | 0.93±0.02 | 0.98±0.00 | 0.91±0.02 | 0.68±0.02 | 0.63±0.05 | 0.68±0.02 | 0.62±0.05 | 0.69±0.02 | 0.64±0.05 | 0.69±0.03 | 0.65±0.06 |
| PatternNet | 0.90±0.05 | 0.79±0.07 | 0.94±0.03 | 0.76±0.05 | 0.76±0.04 | 0.69±0.01 | 0.77±0.04 | 0.70±0.02 | 0.91±0.05 | 0.85±0.07 | 0.91±0.05 | 0.85±0.07 |
| PatternAttribution | 0.99±0.02 | 0.81±0.07 | 0.98±0.03 | 0.80±0.07 | 0.95±0.05 | 0.82±0.05 | 0.95±0.03 | 0.83±0.05 | 0.99±0.01 | 0.93±0.07 | 0.98±0.02 | 0.95±0.04 |
| SHAP | 0.91±0.02 | 0.85±0.02 | 0.88±0.03 | 0.85±0.02 | 0.88±0.03 | 0.82±0.04 | 0.88±0.03 | 0.82±0.04 | 0.87±0.03 | 0.79±0.05 | 0.86±0.02 | 0.82±0.03 |
| Int. Grads. | 0.99±0.00 | 0.96±0.01 | 0.99±0.00 | 0.95±0.01 | 0.98±0.01 | 0.87±0.00 | 0.98±0.01 | 0.88±0.00 | 0.94±0.03 | 0.93±0.03 | 0.91±0.03 | 0.94±0.03 |

Table 1: False-Negative Invariant Earth Mover's Distance Accuracy (FNI-EMDA) shown as mean $\pm$ standard deviation values. The best performing method per row is emboldened, and our proposed importance metrics are denoted by ($\dagger$).

## 4.2 COMPAS Recidivism Risk

COMPAS is a score developed to predict recidivism risk, used to inform bail, sentencing and parole decisions, and is notorious for the presence of racial bias (Angwin et al., 2016; Dressel & Farid, 2018; Tan et al., 2018). Here we investigate the importance of six features, age, race, prior's count, length of stay, sex, and charge degree, on recidivism risk using a publicly available dataset (ProPublica, 2016). Figure 3 shows the learned shape functions from an ensemble of 100 NAMs. Sensitive features like

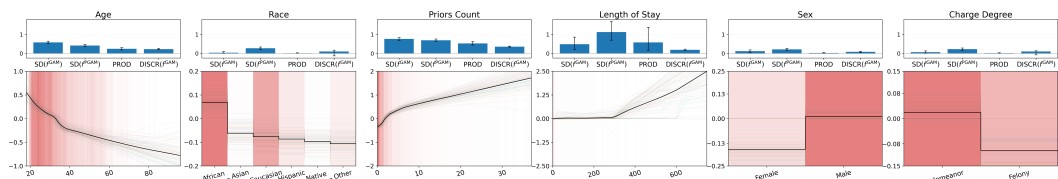

Figure 3: NAM shape functions for six predictors of recidivism risk as estimated by an ensemble of 100 NAMs. Red shading highlights the data density. Blue bar plots mark global importance according to four metrics: NAM scale, SD($f^{GAM}$), PatternGAM scale, SD($f^{PGAM}$), the product of both (PROD), and univariate discriminability (DISCR). In this example, all six features show (marginal but) significant dependencies to recidivism risk, while the fitted multivariate NAM primarily focuses on age, length of stay, and priors count.

race and sex, as well as charge degree are of low importance for the multivariate NAM prediction

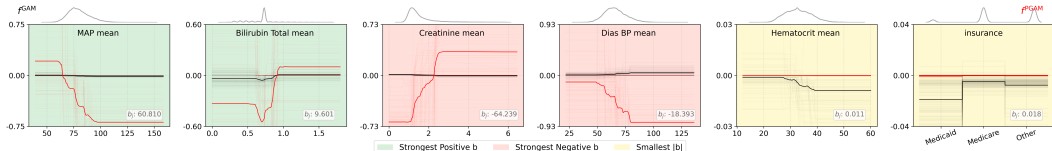

Figure 4: Learned shape functions for an ensemble of 100 NAMs trained on MIMIC-IV data to predict mortality within the first 24 hours of hospital admission, versus PatternGAM shape functions. Green-shaded plots show features with highest univariate informativeness with respect to mortality relative to their use in the NAM. Red-shaded plots depict features entering the model with a polarity that opposes their association with mortality, suggesting negative suppression effects. Yellow-shaded features are features used by the NAM for variance reduction without being predictive for mortality. These classical suppressors are nullified by PatternGAM.

as evidenced by low values of the $\mathrm{SD}(f^{\mathrm{GAM}})$ metric, despite displaying modest but significant correlation to the target according to the $\mathrm{DISCR}(f^{\mathrm{GAM}})$ metric and univariate PatternGAM analysis. We may speculate that these latter weak statistical associations could be due to a confounding effects such as selection biases affecting the dataset's composition rather than these features being truly causal for recidivism. Consistent with this two-sided interpretation, we observe that the PROD metric, quantifying to what extent features are simultaneously informative and used by the model, attains near-zero values for race, sex, and charge degree. These results suggest that the multivariate NAM can achieve its predictive accuracy largely without relying on these sensitive attributes, which could be considered a desirable outcome. This may allow for more nuanced bias mitigating schemes than simply removing features based on raw correlations or a-priori assumptions.

### 4.3   MIMIC-IV Mortality Prediction

Finally, we evaluate additive models on in-hospital mortality prediction using the MIMIC-IV v2.0 dataset (Johnson et al., 2023a,b) hosted on the Physionet platform. The task is formulated as binary classification, where the target indicates whether a patient died during the hospital admission within the first $T = 24$ hours of admission. The full details of which features are used and the steps for data preprocessing are shown in Appendix Section F.

Figure 4 shows the learned NAM versus PatternGAM-scaled shape functions for a subset of features studied, with the full results shown in Appendix Figure 10. The green, yellow, and red shaded features are selected according to the strongest positive, strongest negative, and smallest absolute $b^{\mathrm{PGAM}}$ coefficients respectively, fit through the post-hoc univariate PatternGAM models $f^{\mathrm{PGAM}}$. The ensemble of 100 NAMs achieves an AUROC of $0.821$ with zero feature pairs, which has a minor increase in performance (to AUROC $= 0.824$) with 64 pairs chosen by the FAST algorithm – as such, we choose the former, simpler, model for further investigation.

MAP and Bilirubin mean play a minor role in predicting mortality within the multivariate model, however when modelled univariately, PatternGAM finds these features to be strongly informative. This could indicate that these features contain less direct information about the target, presumably through unobserved confounding or indirect causation, than other features, which are then preferred by the model. It is also noted that the scale of PatternGAM shape functions is generally elevated compared to GAM functions, as the latter distributes importance over a maximum of (here) 34 features, while the former must use single features for approximating the same target. Interestingly, Dias BP mean shows a negative suppression effect (Darlington, 1968). While intermediate and high Dias BP mean values are negatively associated with mortality, these values enter the NAM's prediction positively, likely for the purpose of removing BP-related variance from other informative features. Finally, the yellow-shaded Hematocrit mean and insurance features, while playing a role in the NAM, are nullified in the PatternGAM shape function $f^{\mathrm{PGAM}}$, resembling classical suppressors.

## 5   Discussion

Additive models such as GAMs and NAMs perform on par with deep neural networks in a large range of tasks. The purported gain in interpretability of these 'glassbox' methods is, however, elusive. The common interpretation of shape functions as 'highlighting features that are associated with the

predicted target' is often unjustified for dependent data in the presence of suppressors. Likewise, the strengths and signs of these shape functions cannot be unambiguously related to the prediction target. Here we introduce two general approaches, PatternQLR and PatternGAM, which rely on univariate associations with the target when assigning importance to features and shape functions, thus providing information that is typically desired by developers and users of XAI, and that can facilitate correct decision making in downstream tasks. Conversely, we exposed misinterpretations of conventional additive models both using quantitative ground-truth experiments and real-world analyses. It is important to note, though, that PatternQLR and PatternGAM function as a post-hoc adjustments for explanations – they are not designed to be predictive models in themselves or for altering a model's underlying predictive mechanisms.

PatternGAM effectively ignores suppressor variables, avoiding their misinterpretation as being target-informative. Combined with classical GAM shape function analysis, PatternGAM can enable users to identify suppressors as features used by the model to reduce predictive variance rather than enabling the prediction itself. In the present MIMIC analyses, this has led to the discovery of negative suppression, where the correlation of certain clinical features with mortality is the opposite of what classical GAM analysis would suggest.

Misinterpretations of suppression effects are widespread in the XAI community and can often be attributed to a lack of understanding of how multivariate models operate, neglecting that ML models depend on the causal structure of their training data, and may need to assign weight to non-informative features to remove variance. Notably, such misinterpretations can lead to ineffective or even harmful actions downstream. Agarwal et al. state that the inclusion of sensitive features such as race and sex may lead to less fair models for making bail decisions; thus advocating for exclusion. Similarly, Wang et al. (2021) present a GAM analysis suggesting that patients with asthma are prone to have lower risk of dying from pneumonia. This interpretation prompts the authors to suggest manual adjustments to the GAM shape function to 'correct' this effect.

In contrast, we promote a two-step approach, in which a GAM is first used to learn potentially complex non-linear features, after which PatternGAM is applied post-hoc to assess the contextual importance of these feature through their statistical relationship to the target. In the recidivism example, this approach would flag several sensitive attributes as important, thus potentially biased, but would not necessarily mandate exclusion of these attributes due to their insignificance for the model. In the pneumonia example, PatternGAM would offer a straightforward way to assess whether the role of asthma lies indeed in contributing target-related information or whether asthma is merely used as a suppressor by the model. In the latter case, manual adjustment would not be suitable, likely leading to performance degradation of a potentially optimal model, or worse.

Several modelling choices and technical intricacies of PatternQLR or PatternGAM warrant deeper discussion. First, univariate shape functions can be fitted to either predicted or observed labels using either linear rescaling or complete non-linear re-estimation of the GAM shape functions. These choices offer different solutions within the spectrum between purely data- and purely model-centric explanations. The proposed approach of linearly rescaling GAM functions to fit true labels thereby offers interpretability in terms of the actual building blocks used by the model, while possessing the SAP. Second, statistical guarantees such as for the SAP are provided here only for univariate PatternGAM and PatternQLR attributions. These results can be extended to bivariate features in the sense that feature pairs acting jointly as suppressors are correctly rejected by all proposed methods. Consider the structural model $x_1 = y + n_1 n_2, x_2 = n_1, x_3 = n_2$, where $n_{1/2} \sim \mathcal{N}(0,1)$. The non-zero shape functions of the optimal GAM are $f_1^{\text{GAM}}(x_1) = x_1$ and $f_{2,3}^{\text{GAM}}(x_2, x_3) = -x_2 x_3 \neq 0$ despite the fact that $x_2$ and $x_3$ neither individually nor jointly have any statistical association to $y$. In contrast, all proposed pattern-based methods would assign no significant importance to $(x_2, x_3)$. For the structural model $x_1 = y/n, x_2 = n, n \sim \mathcal{N}(0,1)$, though, the original QLR and GAM models consist of the single term $f_{1,2}^{\text{QLR/GAM}}(x_1, x_2) = x_1 x_2$, for which the PatternQLR and PatternGAM shape function coefficients evaluate to $b_{1,2}^{\text{PQLR/PGAM}} = 1$; thus, indirectly assigning importance to the suppressor $x_2$. Importantly, though, the IMP_AGG rule prevents that this bivariate importance leaks into the aggregated score for $x_2$. Future work will define strict data-driven notions of (higher order) feature interactions, and propose appropriate metrics to disentangle different types of suppression effects from true interactions within the framework of additive models.

## Acknowledgments and Disclosure of Funding

This result is part of a project that has received funding from the European Research Council (ERC) under the European Union's Horizon 2020 research and innovation programme (Grant No. 758985), and the German Federal Ministry for Economic Affairs and Climate Action (BMWK) within the "Metrology for Artificial Intelligence in Medicine (M4AIM)" program of the "QI-Digital" initiative.

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

# Technical Appendices and Supplementary Material

## A    Theoretical properties of PatternQLR and PatternGAM

**Proof 1 (Theorem 1)** *We show that $x_j \perp\!\!\!\perp y \Rightarrow s_j$ implies no importance.*

1. *From $x_j \perp\!\!\!\perp y$ follows that the optimal univariate LLR models $g(E[y|\mathbf{x}]) = b_i^{PQLR/PGAM} z_i + d_i^{PQLR/PGAM}$ with $z_i^{QLR} = x_j$ and $z_i^{GAM} = f_j(x_j)$ are constant functions of $z_i$. Therefore, the fitted $b_i^{PQLR/PGAM} \to 0$ for $N \to \infty$.*

2. *a) Similarly, the original PatternQLR and PatternGAM models $a_i^{PQLR/PGAM} y + v_i^{PQLR/PGAM} = z_i^{QLR/GAM}$ are constant functions of $y$ if the true labels $y$ rather than $g(\hat{y})$ are regressed onto $z_i$ using OLS. Also in this case, $a_i^{PQLR/PGAM} \propto Cov[z_i^{QLR/GAM}, y] \to 0$ for $N \to \infty$.*

   *b) For specific models, the predicted labels $\hat{y}$ are linear functions of the input data (that is the identity function) and can be shown to be uncorrelated to $x_j$. This is the case for OLS regression models and unregularised linear discriminant analysis acting on non-linear features $\mathbf{z}$ (Haufe et al., 2014). In these cases, Pattern coefficients estimated via $a_i^{PQLR/PGAM} \propto Cov[z_i^{QLR/GAM}, \hat{y}] \to 0$.*

3. *From $b_i^{PQLR/PGAM} \to 0$ it follows that $f_i^{PQLR/PGAM} \to 0$.*

4. *Discriminability metrics such as DISCR measure nothing but the statistical association between $y$ and $f_j(x_j)$ and therefore possess the SAP by construction.*

**Proof 2 (Theorem 2)** *Consider the linear model Eq. (1). The optimal GAM shape function for $x_2$ is given by $f_2^{GAM} = w_2 x_2$ with $w_2 = -(1 + (cs_1/s_2)^2)^{-1/2} cs_1 s_2^{-1}$, which is non-zero almost everywhere for $N \to \infty$ if $c \neq 0$.*

## B    Code, data availability, and computational resources

### B.1    Code implementation and availability

Anonymised code is available at `https://github.com/braindatalab/pattern-gam` with the `GPL-3.0` license.

The `nam` subfolder is adapted from the official Pytorch implementation of Agarwal et al. (2021), available at `https://github.com/lemeln/nam`.

The `xai_tris` subfolder is adapted from the XAI-TRIS implementation available at `https://github.com/braindatalab/xai-tris` with the `GPL-3.0` license.

### B.2    Data availability

The XAI-TRIS datasets (Clark et al., 2024b) are available to generate at `https://github.com/braindatalab/xai-tris` with the `GPL-3.0` license, with fixed random seeds used for reproducibility.

The COMPAS Recidivism data (ProPublica, 2016) are available at `https://github.com/propublica/compas-analysis`, and we provide the specific data file used `recid.data` in the anonymised GitHub repository.

The MIMIC-IV dataset (Johnson et al., 2023b) is available via PhysioNet (Johnson et al., 2023a) at `https://physionet.org/content/mimiciv/2.0/`, where training is required on subject handling before access can be granted. We use the `v2.0` version of MIMIC-IV.

### B.3    Computational resources

All experiments are able to be computed on personal devices, where we have used an M4 Pro MacBook Pro laptop for a lot of the prototyping and processing work involved. Due to the nature of

the data and the lightweight models, no individual model takes more than a minute or two to train on such a laptop. Preprocessing the MIMIC-IV data takes slightly longer, however this is within the order of 30 to 60 minutes. For the XAI-TRIS model training line search and explanation calculation, we distribute jobs across a maximum of four NVIDIA A40 GPUs to parallelise the process of using multiple seeds and datasets, however these scripts would likely take around 6 to 18 hours if run locally and sequentially.

## C   False-Negative Invariant Earth Mover's Distance Accuracy (FNI-EMDA)

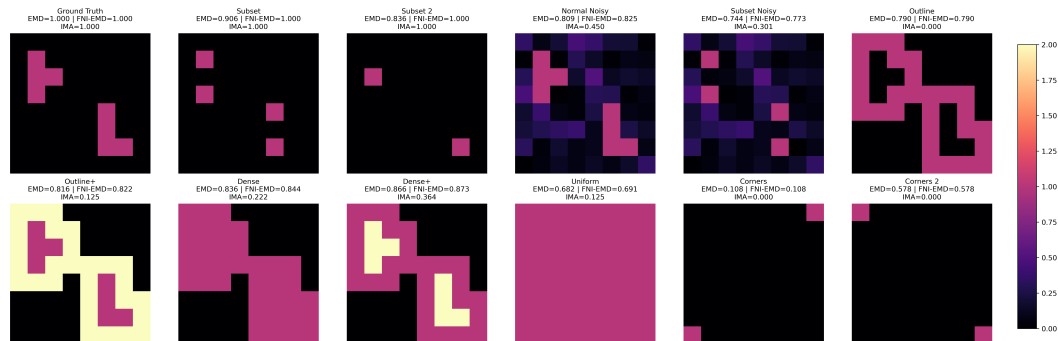

Figure 5: Examples of different types of explanations and their corresponding Earth Mover's Distance (EMD), False-Negative Invariant EMD (FNI-EMD), and Importance Mass Accuracy (IMA) scores when compared to the ground truth. This shows the intuition of why FNI-EMD is beneficial over EMD and IMA, where we can achieve false-negative invariance while preserving a continuous and distance-based measure of explanation correctness. With EMD and FNI-EMD, an explanation outlining the ground truth is not penalised compared to IMA. One thing to note is the greater resolution of scores for this formulation of the EMD-based metrics, due to the ground truth-aware denominator normalisation of the transport cost compared to the original $\delta_{\max}$ formulation proposed (Clark et al., 2024b). This is visible in the 'Corners' explanation, which represents near the lower bound of performance due to attribution as far from the ground truth as possible. Here, our new formulation of the EMD-based metrics has a lower bound of $0.108$, compared to the original formulation's lower bound of $0.473$.

Appendix Figure 5 shows examples of different types of explanations and their corresponding Earth Mover's Distance Accuracy (EMDA), False-Negative Invariant EMD Accuracy (FNI-EMDA), and Importance Mass Accuracy (IMA) scores when compared to the ground truth. This shows the intuition of why FNI-EMDA is beneficial over EMDA and IMA, where we can achieve false-negative invariance while preserving a continuous and distance-based measure of explanation correctness. With EMDA and FNI-EMDA, an explanation outlining the ground truth is not penalised compared to IMA. In higher-resolution imaging tasks, the discrepancy between EMD-based metrics and IMA for outline-style explanations may grow larger, as the entire contiguous mass of signal would have to be highlighted as important for IMA to not penalise the explanation. One thing to note is the narrower range of scores for the original formulation of the EMD-based metrics, where the 'Corners' explanation represents near the lower bound of performance at $0.473$ due to attribution as far from the ground truth as possible. With our ground truth-aware formulation, this lower bound is reduced to $0.108$, presenting a higher resolution of values possible.

# D XAI-TRIS Implementation

## D.1 Data Generation

Following from Clark et al. (2024b), we use the $8 \times 8$ variant, where each dataset consists of images of size, formulated as $D = \{(x^{(n)}, y^{(n)})\}_{n=1}^{N}$. This contains independent and identically distributed observations $(x^{(n)} \in \mathbb{R}^D, y^{(n)} \in \{0, 1\})_{n=1}^{N}$ with $N = 10000$ and dimensionality $D = 64$. The entities $x^{(n)}$ and $y^{(n)}$ represent instances of the stochastic variables $X$ and $Y$, governed by a joint probability density function $p_{X,Y}(x, y)$. In each defined scenario, the instance $x^{(n)}$ is synthesised by integrating a signal pattern $a^{(n)} \in \mathbb{R}^D$, encapsulating the ground truth features for explanations, with background noise $\eta^{(n)} \in \mathbb{R}^D$. The noise component $\eta^{(n)}$, indicative of a non-informative background, is drawn from a multivariate normal distribution $\mathcal{N}(0, I_D)$, resulting in zero mean and an identity covariance $I_D$ white Gaussian noise. This setup is designated as the WHITE scenario. In each classification task, an alternate background, CORR, is specified where a two-dimensional Gaussian spatial smoothing filter $G : \mathbb{R}^D \to \mathbb{R}^D$ modifies the noise element $\eta^{(n)}$, with the smoothing parameter (spatial standard deviation of the Gaussian) set to $\sigma_{\text{smooth}} = 3$. As discussed later in Section D.1.3, the correlated noise backgrounds are where suppressors emerge, where the strength of $\sigma_{\text{smooth}}$ determines the strength of suppression in the 'support' or distance of correlation between pixels. Here, we also propose an explicit form of a suppressor in the optional distractor pattern $d^{(n)} \in \mathbb{R}^D$, where this form of background component contains no class-related information, but provides a more explicit overlap with ground truth features $a^{(n)}$ than the correlated noise background. When this explicit distractor is used, we designate the specify the scenario as DIST WHITE or DIST CORR depending on the other background component used.

In Clark et al.'s analysis, a scenario is also considered where the signal pattern $a^{(n)}$ undergoes a random spatial rigid body transformation (involving translation and rotation of the tetromino; shortened to RIGID) $R^{(n)} : \mathbb{R}^D \to \mathbb{R}^D$. We do not consider this scenario here, as we are looking to provide global explanations, and the non-fixed nature of the signal in this case means that there is no single global explanation. In all other scenarios, the identity transformation is utilised, such that $R^{(n)} \circ a^{(n)} = a^{(n)}$. The transformed signal, distractor, and noise components, $(R^{(n)} \circ a^{(n)})$ and $(G \circ \eta^{(n)})$, are horizontally concatenated into matrices $A = \{(R^{(1)} \circ a^{(1)}), \ldots, (R^{(N)} \circ a^{(N)})\}$, $D = \{(d^{(1)}), \ldots, (d^{(N)})\}$, and $E = \{(G \circ \eta^{(1)}), \ldots, (G \circ \eta^{(N)})\}$. The three components are then normalised by the Frobenius norms of $A$, $D$ and $E$: $(R^{(n)} \circ a^{(n)}) \leftarrow (R^{(n)} \circ a^{(n)})/\|A\|_F$, $(d^{(n)}) \leftarrow (d^{(n)})/\|D\|_F$, and $(G \circ \eta^{(n)}) \leftarrow (G \circ \eta^{(n)})/\|E\|_F$, where the Frobenius norm of a matrix $M$ is defined as $\|M\|_F := \left(\sum_{n=1}^{N} \sum_{d=1}^{D} (m_d^{(n)})^2\right)^{1/2}$. The weighted sum of the signal, distractor, and background components is computed, where the scalar parameters $\alpha_a, \alpha_d, \alpha_\eta \in [0, 1]$ determines the signal-to-noise ratio (SNR), such that $\alpha_a + \alpha_d + \alpha_\eta = 1$. This forms two generative models, which combine these components either additively or multiplicatively. For data generated through either process, each sample $x^{(n)} \in \mathbb{R}^D$ is scaled to the range $[-1, 1]^D$, such that $x^{(n)} \leftarrow x^{(n)}/\max |x^{(n)}|$, where $\max |x^{(n)}|$ denotes the maximum absolute value of the sample $x^{(n)}$.

### D.1.1 Additive Generation

The data generation process for the $n$-th sample is defined as

$$x^{(n)} = \alpha_a(R^{(n)} \circ a^{(n)}) + \alpha_d d^{(n)} + \alpha_\eta(G \circ \eta^{(n)}), \tag{7}$$

where the signal pattern $a^{(n)} \in \mathbb{R}^D$ varies, embodying tetromino shapes based on the binary class label $y^{(n)}$ which is distributed according to a Bernoulli process with a success probability of 0.5. When the distractor pattern is used, $\alpha_d, \alpha_\eta = (1 - \alpha_a)/2$ for simplicity of parameterisation, however an imbalanced weighting of the two components can be used as long as $\alpha_a + \alpha_d + \alpha_\eta = 1$. If the distractor pattern is not used, $d^{(n)} = \mathbf{0}$. The formulation with an explicit distractor is akin to the data generation process seen in Wilming et al. (2022).

### D.1.2 Multiplicative Generation

The sample-wise data generation process is defined as

$$x^{(n)} = \left(1 - \alpha_a \left(R^{(n)} \circ a^{(n)}\right)\right) \left(1 - \alpha_d d^{(n)}\right) \left(G \circ \eta^{(n)}\right), \tag{8}$$

where $a^{(n)}, d^{(n)}, \eta^{(n)}, R^{(n)}$, and $G$ are defined as previously stated. If the distractor pattern is not used in the final scenario, $d^{(n)}$ is again $\mathbf{0}$.

### D.1.3 Emergence of suppressors

In the scenarios where background noise is correlated, the explicit distractor pattern is used, or both, the presence of suppressor variables is induced in both the additive and the multiplicative data generation cases. A suppressor is identified as a pixel not part of the foreground $R^{(n)} \circ a^{(n)}$, but is correlated with a foreground pixel through the application of the smoothing operator $G$ or spatial overlap of $a^{(n)}$ with $d^{(n)}$. Drawing on characteristics of suppressor variables previously reported (Conger, 1974; Friedman & Wall, 2005; Haufe et al., 2014), it has been hypothesised and since shown that XAI methods erroneously attribute importance to suppressor features in both linear and non-linear settings (Wilming et al., 2022; Oliveira et al., 2024; Clark et al., 2024b). This misattribution can lead to decreased explanation performance when compared to scenarios involving just white noise backgrounds with no distractors.

### D.1.4 Classification scenarios

We study three distinct classification scenarios that are introduced using tetrominoes (Golomb, 1996), which are geometric shapes consisting of four features. They are then utilised to define each signal pattern $a^{(n)} \in \mathbb{R}^{8 \times 8}$ and distractor pattern $d^{(n)} \in \mathbb{R}^{8 \times 8}$ when used. These tetrominos are used to induce statistical associations between the features and the target in the following classification scenarios.

**Linear (LIN)** In the linear case, the additive generation model from equation (7) is employed, where $R^{(n)}$ represents the identity transformation, combining the pure signal pattern and the Gaussian white noise background additively. T-shaped tetromino patterns $a_T$ and L-shaped tetromino patterns $a_L$ are utilised for signal patterns, positioned near the top-left corner if $y = 0$ and near the bottom-right corner if $y = 1$, respectively, thus constituting the binary classification problem. Each four-pixel pattern is encoded such that for each pixel in the tetromino pattern, positioned at the i-th row and j-th column, $a_{i,j}^{T/L} = 1$, and zero otherwise.

**Multiplicative (MULT)** The multiplicative generation process (8) with signal patterns $a_T$, $a_L$ is defined with the same tetrominoes as in the linear case, while transformation $R^{(n)}$ remains the identity transform. In this scenario, a degree of non-linearity is introduced as the foreground tetromino pattern, when overlaying the background noise, is reduced towards zero. Therefore, values either increase or decrease depending on their original sign. The complexity introduced by the non-linearity renders linear classifiers unable to solve this classification problem effectively (Clark et al., 2024b). This configuration is meant to evaluate how different machine learning methods can adjust to and manage intricate, interconnected data presentations that are not linear.

**Exclusive or (XOR)** In the XOR configuration, an additive challenge is presented where both tetromino variants, denoted as $a^{T/L}$, are utilised in each sample, with the transformation $R^{(n)}$ maintaining its role as the identity transform. Within this setup, the class membership is defined such that for the first class (where $y = 0$), a combination of both tetromino shapes is superimposed on the image background, either in a positive or negative overlay, expressed as $a^{XOR++} = a^T + a^L$ and $a^{XOR--} = -a^T - a^L$. Conversely, for the second class (where $y = 1$), the tetromino shapes are displayed in a contrasting manner; one shape is overlaid positively, and the other negatively, denoted as $a^{XOR+-} = a^T - a^L$ and $a^{XOR-+} = -a^T + a^L$. This ensures that all four XOR configurations are represented with equal frequency within the dataset.

In all cases, if the distractor pattern $d$ is used, this represents four $2 \times 2$-px squares in the results of Figure 8 as well as Tables 1, 3, and 4. In the demonstrative example of Figure 1, this instead

represents four $2 \times 3$-px blocks so that the distractors fully overlap the signal. In both cases, each distractor tetromino takes a random sign $+/-$ in each sample, however the strength of suppression could be increased by correlating the signs between each distractor tetromino (i.e., the two left distractor tetrominoes have opposite signs, and the two on the right have opposite signs). Examples of the generated data can be seen in Appendix Figure 6, using the four $2 \times 2$-px formulation of distractor patterns.

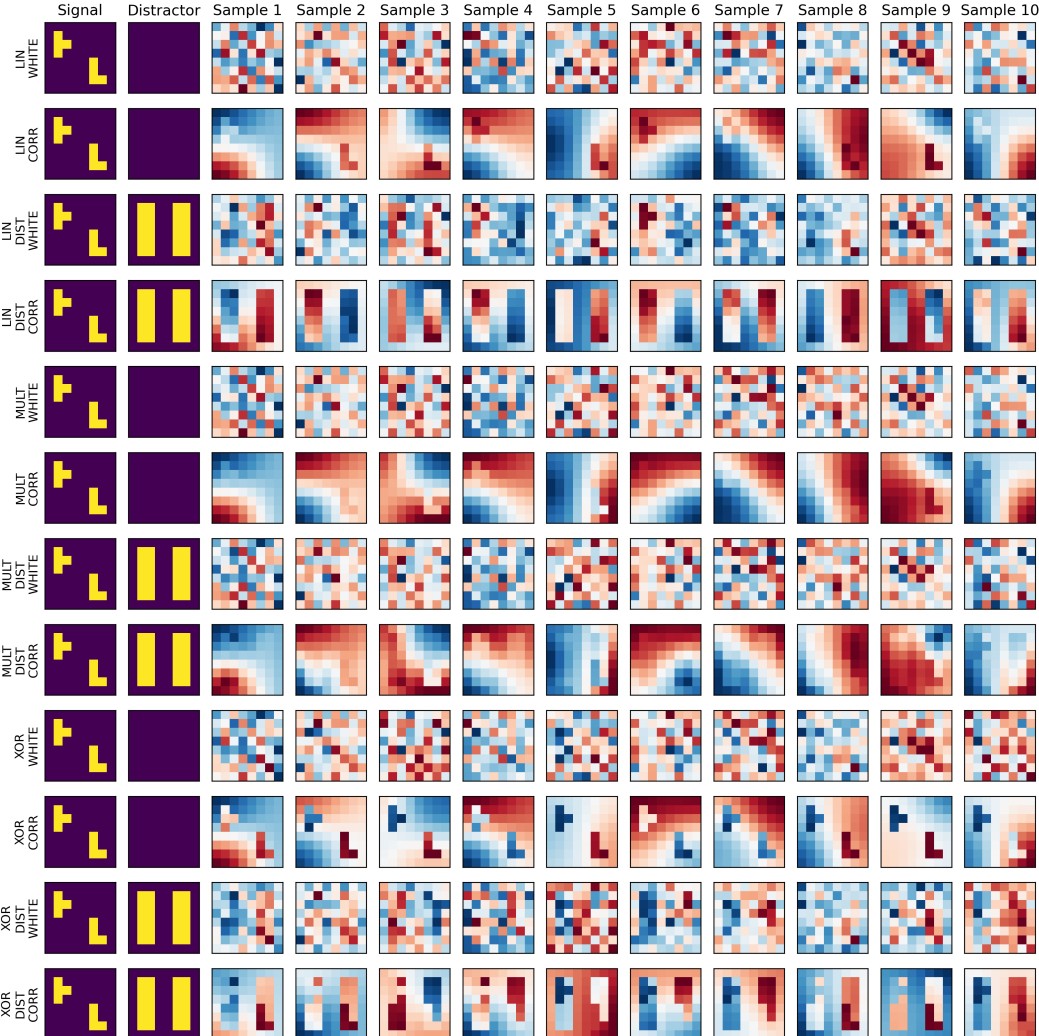

Figure 6: Examples of the data generated for the XAI-TRIS datasets (Clark et al., 2024b), with the adaptation of static distractor patterns as an alternative and explicit form of suppressor variable compared to the correlated noise backgrounds used by the original authors.

## D.2  Defining Ground-Truth Feature Importance

Following Clark et al. (2024a), we can formalise the ground truth sets for significant pixels as follows. For the LIN and MULT scenarios, all pixels belonging to the (smoothed) T and L shaped tetrominoes have a direct statistical association to the target, whereas there are no bivariate feature interactions by construction; therefore

$$\mathcal{A}^+_{\text{LIN, MULT}} := \{j | \left( R^{(n)} \circ \mathbf{a}^{(n)} \right)_j \neq 0, \ j \in \{1, \ldots, 64\}\}, \quad \mathcal{I}^+_{\text{LIN, MULT}} := \emptyset . \tag{9}$$

Note that in line with the definition of importance through the SAP, pixels of both tetrominoes form the ground-truth, even if in the LIN and MULT cases, only one of the tetrominoes is present in each individual image.

In the XOR scenario, no individual pixel has a marginal dependence on the target; however, all pairs of pixels where one pixel is part of the L-shaped tetromino and the other one is part of the T-shaped tetromino form an interaction with the target. Therefore we have

$$\mathcal{A}_{\text{XOR}}^{+} := \emptyset \ , \ \mathcal{I}_{\text{XOR}}^{+} := \{(j,k)| \left( R^{(n)} \circ \mathbf{a}^{T^{(n)}} \right)_j \neq 0 \land \left( R^{(n)} \circ \mathbf{a}^{L^{(n)}} \right)_k \neq 0, \ j,k \in \{1, \ldots, 64\}\} \ . \tag{10}$$

The aggregated input-space ground truth is identical in all scenarios:

$$\mathcal{T}_{\text{LIN, MULT, XOR}}^{+} \equiv \mathcal{A}_{\text{LIN, MULT}}^{+} \equiv \{j \mid \exists k \ (j,k) \in \mathcal{I}_{\text{XOR}}^{+}\} \ . \tag{11}$$

### D.3 Classifiers

Here we outline the parameterisation of each classifier used for subsequent training. The XAI-TRIS library outputs the data pre-split three-fold into training, validation, and test data, defaulting to a $90/5/5$ split respectively. However, all models except the MLP do not take the validation data as input, with the NAM and EBM implementations performing a validation split internally. As such, we concatenate the training and validation data as an input to all models other than the MLP. For the XOR scenarios with the NAM and EBM models, pairwise interactions are first identified using the FAST algorithm (Lou et al., 2013) with 128 interactions.

**Neural Additive Model (NAM)**    We utilise the official PyTorch implementation of NAMs[2]. We instantiate each feature network for both main effects $f_i$ and interaction subnets $f_{ij}$ as Multi-Layer Perceptrons (MLPs) with hidden units $[16, 16, 16]$, taking either one or two features as inputs, respectively. The Adam optimiser is used with the default learning rate of $0.02082$ and with a binary cross-entropy (BCE) loss function, training the model over a maximum of 100 epochs with a patience of 50 epochs. Output penalisation with the default value of $0.2078$ is used to regularise smaller outputs of each subnetwork, similar to ridge regression.

**Quadratic Logistic Regression (QLR)**    We implement a Logistic Regression model from `scikit-learn` with the quadratic feature expansion mentioned in Section 3. No penalty is applied and the intercept is not fitted.

**Multi-Layer Perceptron (MLP)**    A standard MLP is trained using PyTorch. The network architecture consists of an input layer, followed by three ReLU-activated hidden layers of size [32,16,8] and an output layer. The model is trained for a maximum of 100 epochs with a batch size of $64$, using the Adam optimiser with a learning rate of $1e-3$ and a BCE loss function. Early stopping is implemented with a patience of 50 epochs based on the validation loss. The model with the best validation loss is stored and used for evaluation.

**Kernel Support Vector Machine (kSVM)**    We use a Support Vector Classifier from `scikit-learn` with a precomputed Radial Basis Function (RBF) kernel, calculated between all training samples. The parameter $\gamma = 1/(d \times Var(\mathbf{X}))$ defines the distance of influence of a single training example, for $d$ the number of features and $Var(\mathbf{X})$ the sample variance. This can be seen as the inverse of the radius of influence of samples selected by the model as support vectors. For prediction on the test set, the kernel is computed between each test sample and all training samples $K_{\text{test, train}}$.

**Explainable Boosting Machine (EBM)**    Finally, we use the `ExplainableBoostingClassifier` from the `interpret` library. All parameters are kept at their default values.

### D.4 Training

To find the optimal signal-to-noise ratio (SNR) parameterisation, we take a line search across 10 values between zero and one, where the results can be seen in Appendix Figure 7. After this, we

---

[2]`https://github.com/lemeln/nam`

| Scenario | $\alpha_a$ | NAM | QLR | MLP | Kernel SVM | EBM |
|---|---|---|---|---|---|---|
| LIN WHITE | 0.20 | $0.87 \pm 0.02$ | $0.88 \pm 0.02$ | $0.91 \pm 0.01$ | $0.92 \pm 0.01$ | $0.92 \pm 0.01$ |
| LIN CORR | 0.10 | $0.90 \pm 0.06$ | $1.00 \pm 0.00$ | $1.00 \pm 0.00$ | $1.00 \pm 0.00$ | $1.00 \pm 0.00$ |
| LIN DIST WHITE | 0.20 | $0.95 \pm 0.03$ | $0.98 \pm 0.00$ | $0.98 \pm 0.00$ | $0.99 \pm 0.01$ | $0.99 \pm 0.00$ |
| LIN DIST CORR | 0.05 | $0.77 \pm 0.06$ | $1.00 \pm 0.00$ | $1.00 \pm 0.00$ | $1.00 \pm 0.00$ | $1.00 \pm 0.00$ |
| MULT WHITE | 0.60 | $0.79 \pm 0.05$ | $0.90 \pm 0.02$ | $0.93 \pm 0.01$ | $0.84 \pm 0.02$ | $0.95 \pm 0.01$ |
| MULT CORR | 0.40 | $0.81 \pm 0.04$ | $0.99 \pm 0.00$ | $1.00 \pm 0.00$ | $0.96 \pm 0.01$ | $1.00 \pm 0.00$ |
| MULT DIST WHITE | 0.60 | $0.79 \pm 0.05$ | $0.89 \pm 0.02$ | $0.92 \pm 0.01$ | $0.83 \pm 0.02$ | $0.95 \pm 0.01$ |
| MULT DIST CORR | 0.40 | $0.76 \pm 0.04$ | $0.99 \pm 0.01$ | $1.00 \pm 0.01$ | $0.97 \pm 0.02$ | $1.00 \pm 0.00$ |
| XOR WHITE | 0.30 | $0.95 \pm 0.01$ | $0.98 \pm 0.00$ | $0.98 \pm 0.00$ | $0.98 \pm 0.00$ | $0.98 \pm 0.01$ |
| XOR CORR | 0.30 | $0.93 \pm 0.03$ | $1.00 \pm 0.00$ | $1.00 \pm 0.00$ | $1.00 \pm 0.00$ | $0.99 \pm 0.00$ |
| XOR DIST WHITE | 0.20 | $0.91 \pm 0.01$ | $0.94 \pm 0.01$ | $0.96 \pm 0.01$ | $0.95 \pm 0.01$ | $0.95 \pm 0.01$ |
| XOR DIST CORR | 0.20 | $0.91 \pm 0.03$ | $1.00 \pm 0.00$ | $1.00 \pm 0.00$ | $1.00 \pm 0.00$ | $0.98 \pm 0.01$ |

Table 2: XAI-TRIS model accuracy (mean $\pm$ standard deviation) across scenarios and backgrounds for the chosen signal-to-noise ratios (SNRs), shown by the signal parameter $\alpha_a$.

select SNR values for each such that the classification accuracy of all models on the test dataset is at or above $80\%$. The final model accuracies and chosen SNRs (parametrised by the signal strength weighting $\alpha_a$) are shown in Appendix Table 2. As mentioned in the main text, noise strength is set to $\alpha_\eta = 1 - \alpha_a$ when no distractor is present, and $\alpha_d, \alpha_\eta = (1 - \alpha_a)/2$ when distractors are present.

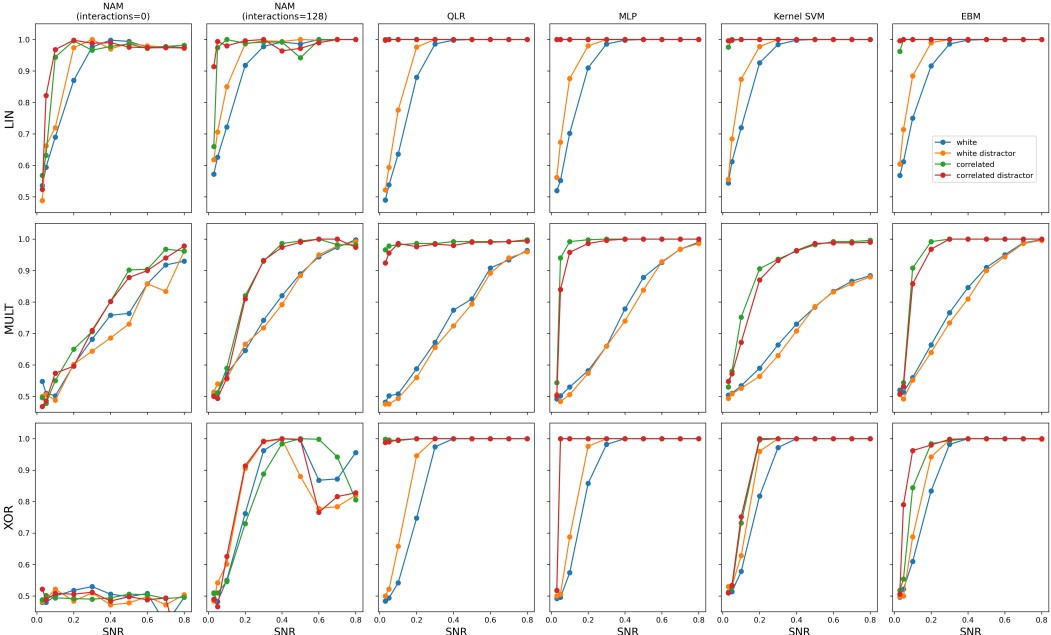

Figure 7: XAI-TRIS model accuracy results for a line search of signal-to-noise ratios (SNRs) for each classification scenario, separated by scenario and model type. Data with suppressor variables present (either as explicit distractor patterns, correlated background noise, or both) results in performant models at significantly lower SNR values.

## D.5 Results

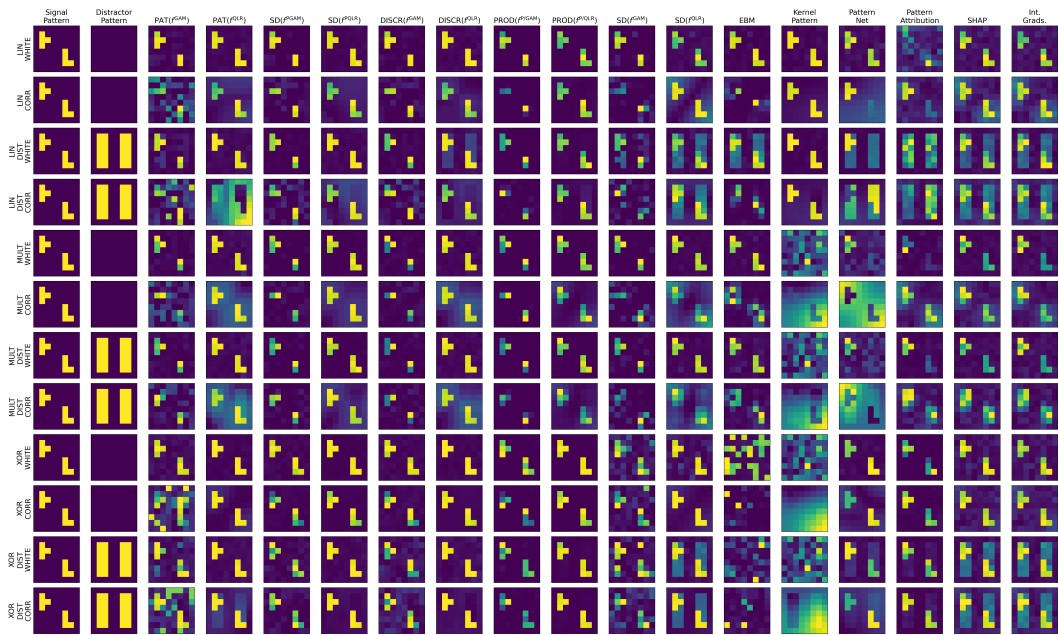

Figure 8: Global heatmaps for XAI-TRIS scenarios across all XAI methods tested. While most methods show at least some attribution to the signal tetrominos, the EBM () and Kernel Pattern (Zhang et al., 2024) struggle, particularly with correlated noise scenarios. PatternNet and PatternAttribution (Kindermans et al., 2019), SHAP (Lundberg et al., 2020-01), and Integrated Gradients (Sundararajan et al., 2017), are all implemented for the Multi-Layer Perceptron (MLP) model, and tend to also attribute strong importance to distractor features directly neighbouring the ground truth signal. All methods proposed in this paper perform well, as also evidenced by the quantitative results of Section 4.1, with no single 'best' method being immediately noticeable qualitatively.

| | LIN | | | | MULT | | | | XOR | | | |
|---|---|---|---|---|---|---|---|---|---|---|---|---|
| Method | WHITE | CORR | DIST WHITE | DIST CORR | WHITE | CORR | DIST WHITE | DIST CORR | WHITE | CORR | DIST WHITE | DIST CORR |
| PAT($f^{GAM}$)† | 0.86 ± 0.02 | 0.40 ± 0.10 | 0.89 ± 0.01 | 0.36 ± 0.11 | 0.81 ± 0.04 | 0.36 ± 0.11 | 0.82 ± 0.03 | 0.38 ± 0.11 | 0.80 ± 0.02 | 0.50 ± 0.11 | 0.75 ± 0.04 | 0.51 ± 0.13 |
| PAT($f^{QLR}$)† | 0.66 ± 0.01 | 0.49 ± 0.06 | 0.72 ± 0.00 | 0.34 ± 0.03 | 0.67 ± 0.01 | 0.19 ± 0.00 | 0.66 ± 0.00 | 0.18 ± 0.00 | 0.75 ± 0.00 | 0.66 ± 0.04 | 0.72 ± 0.01 | 0.61 ± 0.07 |
| SD($f^{PGAM}$)† | 0.97 ± 0.02 | 0.85 ± 0.06 | 0.99 ± 0.01 | 0.78 ± 0.07 | 0.94 ± 0.03 | 0.84 ± 0.07 | 0.94 ± 0.03 | 0.85 ± 0.07 | 0.91 ± 0.02 | 0.90 ± 0.04 | 0.89 ± 0.03 | 0.91 ± 0.03 |
| SD($f^{PQLR}$)† | 0.75 ± 0.00 | 0.78 ± 0.09 | 0.86 ± 0.00 | 0.59 ± 0.07 | 0.88 ± 0.00 | 0.28 ± 0.00 | 0.88 ± 0.00 | 0.27 ± 0.00 | 0.89 ± 0.00 | 0.94 ± 0.02 | 0.86 ± 0.00 | 0.91 ± 0.02 |
| DISCR($f^{GAM}$)† | 0.90 ± 0.01 | 0.80 ± 0.07 | 0.91 ± 0.01 | 0.71 ± 0.07 | 0.74 ± 0.03 | 0.64 ± 0.10 | 0.75 ± 0.03 | 0.64 ± 0.10 | 0.88 ± 0.01 | 0.88 ± 0.03 | 0.82 ± 0.03 | 0.83 ± 0.05 |
| DISCR($f^{QLR}$)† | 0.71 ± 0.01 | 0.43 ± 0.02 | 0.71 ± 0.01 | 0.39 ± 0.02 | 0.70 ± 0.01 | 0.25 ± 0.00 | 0.70 ± 0.01 | 0.24 ± 0.00 | 0.78 ± 0.01 | 0.83 ± 0.03 | 0.74 ± 0.01 | 0.79 ± 0.03 |
| PROD($f^{P|GAM}$)† | **1.00 ± 0.00** | **0.96 ± 0.03** | **1.00 ± 0.00** | **0.94 ± 0.03** | **0.99 ± 0.01** | **0.95 ± 0.03** | **0.99 ± 0.01** | **0.95 ± 0.03** | **0.99 ± 0.01** | 0.97 ± 0.04 | **0.99 ± 0.01** | **0.99 ± 0.01** |
| PROD($f^{P|QLR}$)† | 0.97 ± 0.00 | **0.96 ± 0.02** | 0.99 ± 0.00 | 0.93 ± 0.02 | **0.99 ± 0.00** | 0.43 ± 0.00 | **0.99 ± 0.00** | 0.45 ± 0.01 | **0.99 ± 0.00** | **1.00 ± 0.00** | **0.99 ± 0.00** | **0.99 ± 0.00** |
| SD($f^{GAM}$) | 0.93 ± 0.04 | 0.48 ± 0.05 | 0.76 ± 0.04 | 0.46 ± 0.04 | 0.90 ± 0.03 | 0.46 ± 0.05 | 0.90 ± 0.04 | 0.48 ± 0.05 | 0.65 ± 0.04 | 0.53 ± 0.08 | 0.71 ± 0.05 | 0.59 ± 0.08 |
| SD($f^{QLR}$) | 0.54 ± 0.01 | 0.49 ± 0.00 | 0.53 ± 0.01 | 0.49 ± 0.00 | 0.71 ± 0.01 | 0.23 ± 0.00 | 0.71 ± 0.01 | 0.24 ± 0.00 | 0.62 ± 0.02 | 0.65 ± 0.01 | 0.50 ± 0.02 | 0.61 ± 0.00 |
| EBM | 0.88 ± 0.01 | 0.57 ± 0.00 | 0.63 ± 0.01 | 0.52 ± 0.00 | 0.88 ± 0.01 | 0.47 ± 0.00 | 0.87 ± 0.01 | 0.46 ± 0.01 | 0.12 ± 0.04 | 0.10 ± 0.10 | 0.14 ± 0.07 | 0.10 ± 0.08 |
| Kernel Pattern | 0.89 ± 0.01 | 0.80 ± 0.07 | 0.93 ± 0.01 | 0.74 ± 0.06 | 0.09 ± 0.02 | 0.09 ± 0.03 | 0.09 ± 0.02 | 0.09 ± 0.03 | 0.12 ± 0.03 | 0.11 ± 0.03 | 0.16 ± 0.04 | 0.14 ± 0.05 |
| PatternNet | 0.71 ± 0.16 | 0.40 ± 0.19 | 0.80 ± 0.11 | 0.32 ± 0.14 | 0.32 ± 0.09 | 0.11 ± 0.03 | 0.34 ± 0.09 | 0.12 ± 0.04 | 0.75 ± 0.15 | 0.55 ± 0.20 | 0.74 ± 0.14 | 0.56 ± 0.21 |
| PatternAttribution | 0.96 ± 0.05 | 0.42 ± 0.17 | 0.93 ± 0.08 | 0.37 ± 0.13 | 0.85 ± 0.12 | 0.37 ± 0.12 | 0.86 ± 0.10 | 0.39 ± 0.12 | 0.98 ± 0.02 | 0.78 ± 0.19 | 0.93 ± 0.06 | 0.82 ± 0.11 |
| SHAP | 0.74 ± 0.06 | 0.45 ± 0.04 | 0.54 ± 0.09 | 0.43 ± 0.03 | 0.65 ± 0.07 | 0.39 ± 0.10 | 0.64 ± 0.07 | 0.39 ± 0.08 | 0.60 ± 0.09 | 0.37 ± 0.09 | 0.49 ± 0.07 | 0.38 ± 0.05 |
| Int. Grads. | 0.97 ± 0.01 | 0.86 ± 0.04 | 0.95 ± 0.01 | 0.80 ± 0.05 | 0.93 ± 0.02 | 0.48 ± 0.00 | 0.93 ± 0.02 | 0.48 ± 0.01 | 0.82 ± 0.08 | 0.76 ± 0.12 | 0.67 ± 0.11 | 0.77 ± 0.12 |

Table 3: Importance Mass Accuracy (IMA) quantitative evaluation results. Values are mean ± standard deviation. Best result per row is emboldened.

# E   COMPAS Recidivism risk

| | LIN | | | | MULT | | | | XOR | | | |
|---|---|---|---|---|---|---|---|---|---|---|---|---|
| Method | WHITE | CORR | DIST WHITE | DIST CORR | WHITE | CORR | DIST WHITE | DIST CORR | WHITE | CORR | DIST WHITE | DIST CORR |
| PAT($f^{GAM}$)† | 0.94 ± 0.01 | 0.78 ± 0.03 | 0.92 ± 0.02 | 0.77 ± 0.05 | 0.91 ± 0.04 | 0.76 ± 0.04 | 0.91 ± 0.03 | 0.76 ± 0.05 | 0.93 ± 0.01 | 0.81 ± 0.04 | 0.90 ± 0.01 | 0.81 ± 0.05 |
| PAT($f^{QLR}$)† | 0.88 ± 0.00 | 0.81 ± 0.03 | 0.89 ± 0.00 | 0.76 ± 0.01 | 0.88 ± 0.00 | 0.71 ± 0.00 | 0.88 ± 0.00 | 0.71 ± 0.00 | 0.91 ± 0.00 | 0.89 ± 0.01 | 0.89 ± 0.00 | 0.87 ± 0.03 |
| SD($f^{PGAM}$)† | 0.97 ± 0.02 | 0.93 ± 0.03 | 0.89 ± 0.03 | **0.89 ± 0.03** | 0.94 ± 0.03 | **0.91 ± 0.03** | 0.93 ± 0.04 | **0.91 ± 0.03** | 0.96 ± 0.01 | 0.93 ± 0.02 | **0.93 ± 0.02** | **0.92 ± 0.03** |
| SD($f^{PQLR}$)† | 0.91 ± 0.00 | 0.92 ± 0.04 | 0.84 ± 0.01 | 0.83 ± 0.02 | 0.96 ± 0.00 | 0.75 ± 0.00 | 0.96 ± 0.00 | 0.75 ± 0.00 | 0.96 ± 0.00 | 0.97 ± 0.01 | 0.87 ± 0.00 | 0.87 ± 0.00 |
| DISCR($f^{GAM}$)† | 0.95 ± 0.02 | 0.91 ± 0.03 | 0.89 ± 0.02 | 0.86 ± 0.03 | 0.89 ± 0.02 | 0.85 ± 0.04 | 0.89 ± 0.02 | 0.85 ± 0.03 | 0.95 ± 0.01 | 0.92 ± 0.02 | 0.90 ± 0.02 | 0.88 ± 0.02 |
| DISCR($f^{QLR}$)† | 0.89 ± 0.00 | 0.81 ± 0.01 | 0.89 ± 0.00 | 0.79 ± 0.01 | 0.89 ± 0.00 | 0.75 ± 0.00 | 0.89 ± 0.00 | 0.75 ± 0.00 | 0.92 ± 0.00 | 0.94 ± 0.01 | 0.89 ± 0.00 | 0.90 ± 0.01 |
| PROD($f^{P/GAM}$)† | 0.96 ± 0.03 | 0.84 ± 0.07 | 0.82 ± 0.05 | 0.73 ± 0.12 | 0.91 ± 0.06 | 0.85 ± 0.05 | 0.91 ± 0.05 | 0.82 ± 0.08 | 0.98 ± 0.01 | 0.89 ± 0.03 | 0.89 ± 0.03 | 0.85 ± 0.03 |
| PROD($f^{P/QLR}$)† | **0.98 ± 0.01** | **0.97 ± 0.02** | 0.82 ± 0.01 | 0.85 ± 0.02 | **0.99 ± 0.00** | 0.81 ± 0.00 | **0.98 ± 0.01** | 0.82 ± 0.00 | **0.99 ± 0.00** | **0.98 ± 0.00** | 0.81 ± 0.01 | 0.83 ± 0.00 |
| SD($f^{GAM}$) | 0.96 ± 0.02 | 0.79 ± 0.07 | 0.89 ± 0.04 | 0.71 ± 0.11 | 0.93 ± 0.03 | 0.81 ± 0.04 | 0.93 ± 0.04 | 0.78 ± 0.06 | 0.86 ± 0.02 | 0.81 ± 0.03 | 0.87 ± 0.02 | 0.82 ± 0.03 |
| SD($f^{QLR}$) | 0.83 ± 0.00 | 0.86 ± 0.00 | 0.85 ± 0.01 | 0.83 ± 0.00 | 0.90 ± 0.00 | 0.74 ± 0.00 | 0.89 ± 0.00 | 0.74 ± 0.00 | 0.86 ± 0.01 | 0.90 ± 0.00 | 0.83 ± 0.01 | 0.88 ± 0.00 |
| EBM | 0.96 ± 0.00 | 0.84 ± 0.01 | 0.90 ± 0.01 | 0.79 ± 0.01 | 0.96 ± 0.00 | 0.88 ± 0.00 | 0.95 ± 0.00 | 0.87 ± 0.00 | 0.65 ± 0.03 | 0.58 ± 0.04 | 0.65 ± 0.03 | 0.61 ± 0.08 |
| Kernel Pattern | 0.96 ± 0.00 | 0.92 ± 0.03 | **0.97 ± 0.00** | **0.89 ± 0.02** | 0.67 ± 0.02 | 0.61 ± 0.04 | 0.67 ± 0.02 | 0.61 ± 0.05 | 0.68 ± 0.02 | 0.62 ± 0.04 | 0.68 ± 0.03 | 0.63 ± 0.05 |
| PatternNet | 0.89 ± 0.06 | 0.78 ± 0.07 | 0.90 ± 0.06 | 0.74 ± 0.05 | 0.73 ± 0.03 | 0.68 ± 0.01 | 0.74 ± 0.03 | 0.68 ± 0.02 | 0.85 ± 0.08 | 0.79 ± 0.08 | 0.85 ± 0.07 | 0.80 ± 0.08 |
| PatternAttribution | 0.94 ± 0.05 | 0.75 ± 0.06 | 0.86 ± 0.07 | 0.75 ± 0.07 | 0.77 ± 0.10 | 0.78 ± 0.05 | 0.76 ± 0.09 | 0.77 ± 0.06 | 0.86 ± 0.08 | 0.82 ± 0.09 | 0.87 ± 0.07 | 0.82 ± 0.08 |
| SHAP | 0.84 ± 0.04 | 0.76 ± 0.07 | 0.82 ± 0.04 | 0.77 ± 0.10 | 0.82 ± 0.04 | 0.78 ± 0.05 | 0.83 ± 0.03 | 0.76 ± 0.05 | 0.83 ± 0.03 | 0.75 ± 0.05 | 0.80 ± 0.04 | 0.76 ± 0.04 |
| Int. Grads. | 0.96 ± 0.04 | 0.94 ± 0.02 | 0.84 ± 0.05 | 0.81 ± 0.04 | 0.92 ± 0.05 | 0.86 ± 0.01 | 0.92 ± 0.05 | 0.82 ± 0.02 | 0.86 ± 0.05 | 0.88 ± 0.06 | 0.84 ± 0.05 | 0.86 ± 0.05 |

Table 4: Earth Mover's Distance (EMD) Accuracy quantitative evaluation results. Values are mean ± standard deviation. Best result per row is emboldened.

| Feature | Corr | MI | HSIC |
|---|---|---|---|
| Age | -0.1928 | 0.0161 | 13.6641 |
| Race | -0.1222 | 0.0042 | 8.5912 |
| Prior Counts | 0.2908 | 0.0592 | 32.0686 |
| Length of Stay | 0.1067 | 0.0187 | 7.8228 |
| Sex | 0.1020 | 0.0053 | 3.1343 |
| Charge Degree | -0.1112 | 0.0062 | 5.5309 |

Table 5: Dependence Scores between COMPAS features and Recidivism risk, shown for Pearson correlation, mutual information (MI) and the Hilbert-Schmidt independence criterion (HSIC).

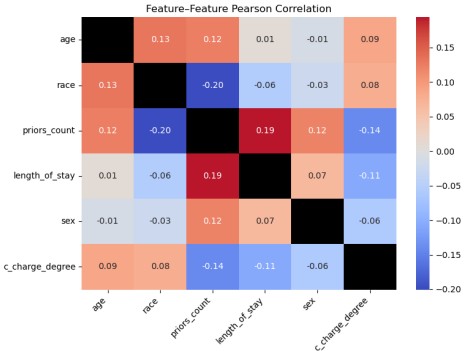

Figure 9: Pearson correlation scores between COMPAS features used to predict recidivism risk.

# F MIMIC-IV Mortality prediction

We construct a cohort for hospital mortality prediction using the MIMIC-IV v2.0 database. Structured tables from the `hosp` and `icu` modules are loaded using pandas, with data filtered to include only hospital admissions with available outcomes.

**Feature extraction.** Outcome labels are derived from the `hospital_expire_flag` field in the `admissions` table. Patient age at admission is computed by combining `anchor_age` and `anchor_year` from the `patients` table with the year of hospital admission. The resulting raw age is capped at 90 to respect MIMIC-IV de-identification procedures. Admissions with implausible or missing age information are excluded.

We extract vital signs and laboratory values from `chartevents` and `labevents`, respectively:

- Vital signs include heart rate, mean arterial pressure (MAP), respiratory rate, SpO2, temperature, and components of the Glasgow Coma Scale (GCS). GCS components (eye, verbal, motor) were summed to create a total GCS score.

- Laboratory variables include sodium, potassium, chloride, bicarbonate, blood urea nitrogen (BUN), creatinine, glucose, white blood cells (WBC), hemoglobin, hematocrit, platelets, total bilirubin, ALT, AST, lactate, pH, and anion gap.

Each event is mapped to a clinical variable using predefined `itemid` dictionaries. For each variable, we compute the mean value over the first $T$ hours after admission, with $T \in \{1, 24, 48, 72\}$.

We show the results for the first $T = 24$ hours after admission. Glasgow Coma Scale (GCS) components are summed to yield a total GCS score.

In addition to time-series features, we include static variables: age, gender, race, insurance type, marital status, primary language, admission type, and source. Race, admission location, and admission type are mapped to coarse categories:

- **Race**: Grouped into White, Black, Hispanic, Asian, and Other/Unknown.
- **Admission Location**: Categorised into Emergency, Transfer In, Referral, Internal/Procedural, and Unknown.
- **Admission Type**: Simplified into Emergency, Urgent, Elective, Observation, and Unknown.

We also compute two aggregate features: the number of diagnoses (`diagnoses_icd`) and the number of prescriptions initiated within the first $T$ hours (`prescriptions`) per hospital admission:

- **Diagnoses count**: Number of unique ICD diagnoses.
- **Prescription count**: Number of prescriptions started within the first $T$ hours.

**Handling missing data and outliers.** We employ a two-stage strategy for missing data. First, rows with missing values in *critical* features are dropped These features are: age, heart rate, MAP, respiratory rate, SpO2, temperature, GCS total, lactate, creatinine. Second, remaining missing values are handled via median imputation for numerical variables and constant-value imputation ('`Unknown`') for categoricals. All categorical variables are then ordinal-encoded with mapping dictionaries saved for interpretability.

Outliers are systematically identified and removed using interquartile range (IQR) filtering. Numerical variables are assessed individually, and observations falling beyond the lower bound ($Q1 - 5 \times \mathrm{IQR}$) or upper bound ($Q3 + 5 \times \mathrm{IQR}$) are excluded. This ensures the robustness of statistical estimates and model stability.

**Cohort finalisation.** After feature construction, we filter to a minimal set of patients with complete outcome and critical data. A stratified group-aware split is applied using a group shuffle split, ensuring that no subject (`subject_id`) appears in both training and test sets.

After preprocessing, the final sample size is $N = 23190$ with 19616 unique patients and 3214 samples of deceased patients, split into training/validation/testing data, leading to a typical class imbalance for medical tasks. The full final list of studied clinical variables and their correlation to the prediction target can be seen in Table 6.

| Feature | Corr Orig. | p-val Orig. | Corr Transf. $f(x)$ | p-val Transf. $f(x)$ |
|---|---|---|---|---|
| pH | -0.077 | <0.001 | 0.092 | <0.001 |
| Marital Status | 0.057 | <0.001 | 0.015 | 0.074 |
| Potassium | 0.062 | <0.001 | 0.057 | <0.001 |
| WBC | 0.078 | <0.001 | 0.087 | <0.001 |
| Heart Rate | 0.080 | <0.001 | 0.074 | <0.001 |
| Creatinine | 0.090 | <0.001 | 0.118 | <0.001 |
| Bilirubin Total | 0.019 | 0.026 | 0.063 | <0.001 |
| Anion Gap | 0.115 | <0.001 | 0.104 | <0.001 |
| Sodium | 0.025 | 0.003 | 0.078 | <0.001 |
| Resp Rate | 0.127 | <0.001 | 0.127 | <0.001 |
| Hematocrit | -0.040 | <0.001 | 0.030 | <0.001 |
| presc count 24h | 0.113 | <0.001 | 0.105 | <0.001 |
| Hemoglobin | -0.064 | <0.001 | 0.054 | <0.001 |
| Age at Admission | 0.161 | <0.001 | 0.161 | <0.001 |
| SpO2 | -0.069 | <0.001 | 0.062 | <0.001 |
| Lactate | 0.152 | <0.001 | 0.120 | <0.001 |
| Admission Type | 0.062 | <0.001 | 0.092 | <0.001 |
| Dias BP | -0.068 | <0.001 | 0.053 | <0.001 |
| Platelets | 0.010 | 0.238 | 0.030 | <0.001 |
| Glucose | 0.033 | <0.001 | 0.038 | <0.001 |
| Sys BP | -0.071 | <0.001 | 0.072 | <0.001 |
| Admission Location | 0.002 | 0.844 | 0.020 | 0.016 |
| MAP | -0.076 | <0.001 | 0.065 | <0.001 |
| ALT | 0.002 | 0.850 | 0.063 | <0.001 |
| Bicarbonate | -0.043 | <0.001 | 0.079 | <0.001 |
| Chloride | -0.004 | 0.638 | 0.033 | <0.001 |
| Temperature | -0.027 | 0.001 | 0.064 | <0.001 |
| Diagnosis count | 0.148 | <0.001 | 0.136 | <0.001 |
| Race | 0.020 | 0.016 | 0.004 | 0.615 |
| Gender | -0.020 | 0.017 | 0.020 | 0.017 |
| Insurance | -0.044 | <0.001 | 0.033 | <0.001 |
| BUN | 0.157 | <0.001 | 0.166 | <0.001 |
| AST | 0.043 | <0.001 | 0.070 | <0.001 |
| GCS Total | -0.235 | <0.001 | 0.189 | <0.001 |

Table 6: MIMIC-IV feature-wise correlations and p-values with the target (Hospital Expire Flag) for mortality prediction $T = 24$ hours after admission. This is shown for the original data, and then for the feature transformed by the learned shape function $f_i(x_i)$ for the given feature. Features like platelets, gender, and marital status have non-zero shape functions but near-zero correlation with the target, and when nullified by PatternGAM, this shows that they may have a suppressive effect on the data. By contrast, features like GCS, age, and BUN have the scale of their importance preserved by PatternGAM, and here we can see that they have stronger correlation with mortality.

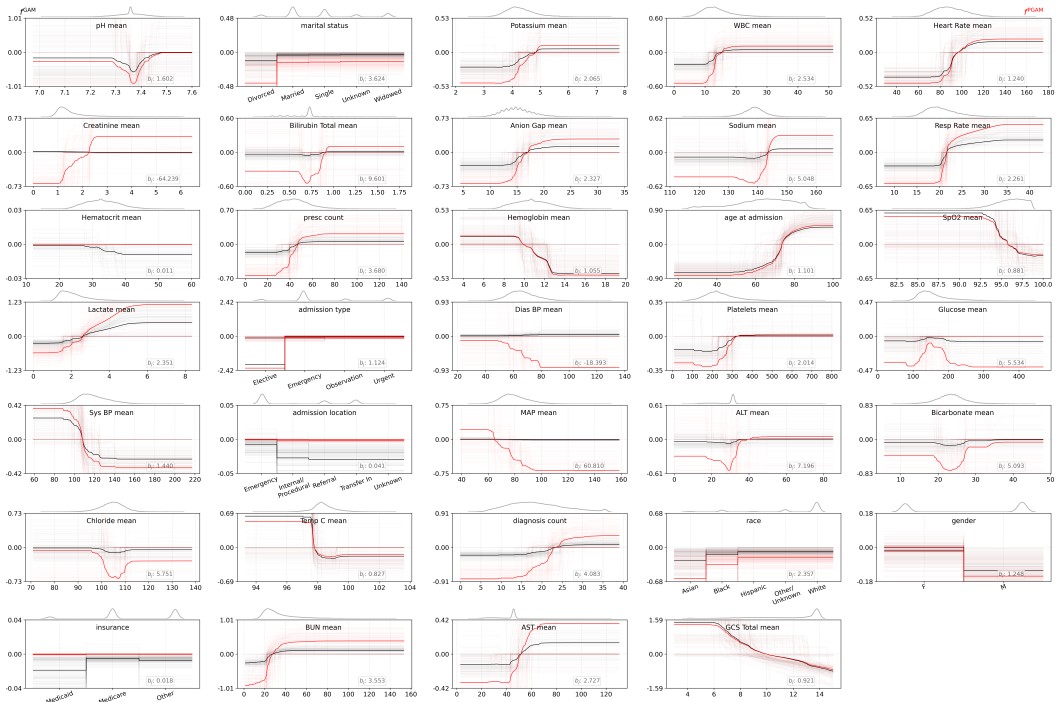

Figure 10: All learned shape functions for an ensemble of 100 NAMs trained on MIMIC-IV mortality data within the first 24 hours of ICU admission, versus PatternGAM shape functions. Data density is shown above each plot, where regions of low density present higher uncertainty in the learned shape functions.

# G Additional Plots

This section shows additional plots referenced in the appendix above that have been added here to not disturb the referencing order given in the submitted main text.

# NeurIPS Paper Checklist

1. **Claims**

   Question: Do the main claims made in the abstract and introduction accurately reflect the paper's contributions and scope?

   Answer: [Yes]

   Justification: The Abstract and Introduction sections reflect the papers contributions and scope, and the introduction states the assumptions and limitations of current work, ending with a dedicated block to state our contributions and limitations of our findings.

   Guidelines:

   - The answer NA means that the abstract and introduction do not include the claims made in the paper.
   - The abstract and/or introduction should clearly state the claims made, including the contributions made in the paper and important assumptions and limitations. A No or NA answer to this question will not be perceived well by the reviewers.
   - The claims made should match theoretical and experimental results, and reflect how much the results can be expected to generalize to other settings.
   - It is fine to include aspirational goals as motivation as long as it is clear that these goals are not attained by the paper.

2. **Limitations**

   Question: Does the paper discuss the limitations of the work performed by the authors?

   Answer: [Yes]

   Justification: The Introduction briefly states the limitations of the work, ending with a dedicated block to state our contributions and limitations of our findings. The Background and Methodology sections list the assumptions made by prior and related work, and how these relate to our contribution. The Discussion section has limitations listed alongside avenues for future work that would overcome these limitations.

   Guidelines:

   - The answer NA means that the paper has no limitation while the answer No means that the paper has limitations, but those are not discussed in the paper.
   - The authors are encouraged to create a separate "Limitations" section in their paper.
   - The paper should point out any strong assumptions and how robust the results are to violations of these assumptions (e.g., independence assumptions, noiseless settings, model well-specification, asymptotic approximations only holding locally). The authors should reflect on how these assumptions might be violated in practice and what the implications would be.
   - The authors should reflect on the scope of the claims made, e.g., if the approach was only tested on a few datasets or with a few runs. In general, empirical results often depend on implicit assumptions, which should be articulated.
   - The authors should reflect on the factors that influence the performance of the approach. For example, a facial recognition algorithm may perform poorly when image resolution is low or images are taken in low lighting. Or a speech-to-text system might not be used reliably to provide closed captions for online lectures because it fails to handle technical jargon.
   - The authors should discuss the computational efficiency of the proposed algorithms and how they scale with dataset size.
   - If applicable, the authors should discuss possible limitations of their approach to address problems of privacy and fairness.
   - While the authors might fear that complete honesty about limitations might be used by reviewers as grounds for rejection, a worse outcome might be that reviewers discover limitations that aren't acknowledged in the paper. The authors should use their best judgment and recognize that individual actions in favor of transparency play an important role in developing norms that preserve the integrity of the community. Reviewers will be specifically instructed to not penalize honesty concerning limitations.

3. **Theory assumptions and proofs**

Question: For each theoretical result, does the paper provide the full set of assumptions and a complete (and correct) proof?

Answer: [Yes]

Justification: In the Background and Methodology sections, we provide the full set of instructions and assumptions on calculating activation patterns for additive models and benchmarking the resulting explanations. The Appendices provide additional details (For example, Appendix Section C provides more details on the benefits and properties of the FNI-EMD metric).

Guidelines:

- The answer NA means that the paper does not include theoretical results.
- All the theorems, formulas, and proofs in the paper should be numbered and cross-referenced.
- All assumptions should be clearly stated or referenced in the statement of any theorems.
- The proofs can either appear in the main paper or the supplemental material, but if they appear in the supplemental material, the authors are encouraged to provide a short proof sketch to provide intuition.
- Inversely, any informal proof provided in the core of the paper should be complemented by formal proofs provided in appendix or supplemental material.
- Theorems and Lemmas that the proof relies upon should be properly referenced.

4. **Experimental result reproducibility**

Question: Does the paper fully disclose all the information needed to reproduce the main experimental results of the paper to the extent that it affects the main claims and/or conclusions of the paper (regardless of whether the code and data are provided or not)?

Answer: [Yes]

Justification: Where possible, we state the key experimental setup details in the main paper, with the Appendices providing the specific and exact instructions. We state the availability of all code to achieve the experimental results in the main text in the form of an anonymised GitHub repository, and point to Appendix Section B which provides additional details on code usage, data availability, computational resources required and the limits thereof. For code, we have used fixed random seeds where possible.

Guidelines:

- The answer NA means that the paper does not include experiments.
- If the paper includes experiments, a No answer to this question will not be perceived well by the reviewers: Making the paper reproducible is important, regardless of whether the code and data are provided or not.
- If the contribution is a dataset and/or model, the authors should describe the steps taken to make their results reproducible or verifiable.
- Depending on the contribution, reproducibility can be accomplished in various ways. For example, if the contribution is a novel architecture, describing the architecture fully might suffice, or if the contribution is a specific model and empirical evaluation, it may be necessary to either make it possible for others to replicate the model with the same dataset, or provide access to the model. In general. releasing code and data is often one good way to accomplish this, but reproducibility can also be provided via detailed instructions for how to replicate the results, access to a hosted model (e.g., in the case of a large language model), releasing of a model checkpoint, or other means that are appropriate to the research performed.
- While NeurIPS does not require releasing code, the conference does require all submissions to provide some reasonable avenue for reproducibility, which may depend on the nature of the contribution. For example
  (a) If the contribution is primarily a new algorithm, the paper should make it clear how to reproduce that algorithm.
  (b) If the contribution is primarily a new model architecture, the paper should describe the architecture clearly and fully.

(c) If the contribution is a new model (e.g., a large language model), then there should either be a way to access this model for reproducing the results or a way to reproduce the model (e.g., with an open-source dataset or instructions for how to construct the dataset).

(d) We recognize that reproducibility may be tricky in some cases, in which case authors are welcome to describe the particular way they provide for reproducibility. In the case of closed-source models, it may be that access to the model is limited in some way (e.g., to registered users), but it should be possible for other researchers to have some path to reproducing or verifying the results.

5. **Open access to data and code**

Question: Does the paper provide open access to the data and code, with sufficient instructions to faithfully reproduce the main experimental results, as described in supplemental material?

Answer: [Yes]

Justification: Similar to the prior question, we state the availability of all code to achieve the experimental results in the main text in the form of an anonymised GitHub repository, and point to Appendix Section B which provides additional details on code usage, data availability, computational resources required and the limits thereof. For code, we have used fixed random seeds where possible. All data is openly available, with only the MIMIC-IV data (Johnson et al., 2023b) requiring the completion of a data usage agreement and a required training course available via Physionet (Johnson et al., 2023a).

Guidelines:

- The answer NA means that paper does not include experiments requiring code.
- Please see the NeurIPS code and data submission guidelines (`https://nips.cc/public/guides/CodeSubmissionPolicy`) for more details.
- While we encourage the release of code and data, we understand that this might not be possible, so "No" is an acceptable answer. Papers cannot be rejected simply for not including code, unless this is central to the contribution (e.g., for a new open-source benchmark).
- The instructions should contain the exact command and environment needed to run to reproduce the results. See the NeurIPS code and data submission guidelines (`https://nips.cc/public/guides/CodeSubmissionPolicy`) for more details.
- The authors should provide instructions on data access and preparation, including how to access the raw data, preprocessed data, intermediate data, and generated data, etc.
- The authors should provide scripts to reproduce all experimental results for the new proposed method and baselines. If only a subset of experiments are reproducible, they should state which ones are omitted from the script and why.
- At submission time, to preserve anonymity, the authors should release anonymized versions (if applicable).
- Providing as much information as possible in supplemental material (appended to the paper) is recommended, but including URLs to data and code is permitted.

6. **Experimental setting/details**

Question: Does the paper specify all the training and test details (e.g., data splits, hyperparameters, how they were chosen, type of optimizer, etc.) necessary to understand the results?

Answer: [Yes]

Justification: Where possible, we have provided details in the main text, with the complete details available in the Appendices. The full code to run all experiments is available with a link stated at the start of the Results section and additional details in Appendix Section B.

Guidelines:

- The answer NA means that the paper does not include experiments.
- The experimental setting should be presented in the core of the paper to a level of detail that is necessary to appreciate the results and make sense of them.

- The full details can be provided either with the code, in appendix, or as supplemental material.

7. **Experiment statistical significance**

   Question: Does the paper report error bars suitably and correctly defined or other appropriate information about the statistical significance of the experiments?

   Answer: [Yes]

   Justification: We provide error bars for the quantitative results of Table 1 in the main text, and for training results that are elaborated in the Appendices where appropriate. Details on the number of experiments/models/datasets run for each are provided in the Appendices along with the full instructions on parameterisation.

   Guidelines:

   - The answer NA means that the paper does not include experiments.
   - The authors should answer "Yes" if the results are accompanied by error bars, confidence intervals, or statistical significance tests, at least for the experiments that support the main claims of the paper.
   - The factors of variability that the error bars are capturing should be clearly stated (for example, train/test split, initialization, random drawing of some parameter, or overall run with given experimental conditions).
   - The method for calculating the error bars should be explained (closed form formula, call to a library function, bootstrap, etc.)
   - The assumptions made should be given (e.g., Normally distributed errors).
   - It should be clear whether the error bar is the standard deviation or the standard error of the mean.
   - It is OK to report 1-sigma error bars, but one should state it. The authors should preferably report a 2-sigma error bar than state that they have a 96% CI, if the hypothesis of Normality of errors is not verified.
   - For asymmetric distributions, the authors should be careful not to show in tables or figures symmetric error bars that would yield results that are out of range (e.g. negative error rates).
   - If error bars are reported in tables or plots, The authors should explain in the text how they were calculated and reference the corresponding figures or tables in the text.

8. **Experiments compute resources**

   Question: For each experiment, does the paper provide sufficient information on the computer resources (type of compute workers, memory, time of execution) needed to reproduce the experiments?

   Answer: [Yes]

   Justification: We provide the details on compute resources in Appendix Section B.

   Guidelines:

   - The answer NA means that the paper does not include experiments.
   - The paper should indicate the type of compute workers CPU or GPU, internal cluster, or cloud provider, including relevant memory and storage.
   - The paper should provide the amount of compute required for each of the individual experimental runs as well as estimate the total compute.
   - The paper should disclose whether the full research project required more compute than the experiments reported in the paper (e.g., preliminary or failed experiments that didn't make it into the paper).

9. **Code of ethics**

   Question: Does the research conducted in the paper conform, in every respect, with the NeurIPS Code of Ethics https://neurips.cc/public/EthicsGuidelines?

   Answer: [Yes]

   Justification: We have reviewed the NeurIPS Code of Ethics and confirm that the research complies with these guidelines in every respect.

Guidelines:

- The answer NA means that the authors have not reviewed the NeurIPS Code of Ethics.
- If the authors answer No, they should explain the special circumstances that require a deviation from the Code of Ethics.
- The authors should make sure to preserve anonymity (e.g., if there is a special consideration due to laws or regulations in their jurisdiction).

10. **Broader impacts**

    Question: Does the paper discuss both potential positive societal impacts and negative societal impacts of the work performed?

    Answer: [Yes]

    Justification: In the introduction we justify why misinterpretation (particularly with respect to suppressor variables being highlighted as important in explanations) is harmful, and why it is necessary to avoid this. Throughout the paper we discuss real world examples in the COMPAS recidivism risk (ProPublica, 2016) and MIMIC-IV datasets (Johnson et al., 2023b). In the Discussion section we revisit the argumentation by other authors in the field related to our work, and the broader impacts of our work.

    Guidelines:

    - The answer NA means that there is no societal impact of the work performed.
    - If the authors answer NA or No, they should explain why their work has no societal impact or why the paper does not address societal impact.
    - Examples of negative societal impacts include potential malicious or unintended uses (e.g., disinformation, generating fake profiles, surveillance), fairness considerations (e.g., deployment of technologies that could make decisions that unfairly impact specific groups), privacy considerations, and security considerations.
    - The conference expects that many papers will be foundational research and not tied to particular applications, let alone deployments. However, if there is a direct path to any negative applications, the authors should point it out. For example, it is legitimate to point out that an improvement in the quality of generative models could be used to generate deepfakes for disinformation. On the other hand, it is not needed to point out that a generic algorithm for optimizing neural networks could enable people to train models that generate Deepfakes faster.
    - The authors should consider possible harms that could arise when the technology is being used as intended and functioning correctly, harms that could arise when the technology is being used as intended but gives incorrect results, and harms following from (intentional or unintentional) misuse of the technology.
    - If there are negative societal impacts, the authors could also discuss possible mitigation strategies (e.g., gated release of models, providing defenses in addition to attacks, mechanisms for monitoring misuse, mechanisms to monitor how a system learns from feedback over time, improving the efficiency and accessibility of ML).

11. **Safeguards**

    Question: Does the paper describe safeguards that have been put in place for responsible release of data or models that have a high risk for misuse (e.g., pretrained language models, image generators, or scraped datasets)?

    Answer: [Yes]

    Justification: We discuss in the results for the COMPAS dataset (ProPublica, 2016) that it is very important to tread carefully in the analysis of biases (there specifically biases against race and sex) and the notion of what 'correcting' such biases entails. Our approach aims to prevent misuse and promote safe and fair usage of machine learning models in the context of interpretability.

    Guidelines:

    - The answer NA means that the paper poses no such risks.
    - Released models that have a high risk for misuse or dual-use should be released with necessary safeguards to allow for controlled use of the model, for example by requiring

that users adhere to usage guidelines or restrictions to access the model or implementing safety filters.

- Datasets that have been scraped from the Internet could pose safety risks. The authors should describe how they avoided releasing unsafe images.
- We recognize that providing effective safeguards is challenging, and many papers do not require this, but we encourage authors to take this into account and make a best faith effort.

12. **Licenses for existing assets**

Question: Are the creators or original owners of assets (e.g., code, data, models), used in the paper, properly credited and are the license and terms of use explicitly mentioned and properly respected?

Answer: [Yes]

Justification: We state the ownership of code and data in Appendix Section B, with subsequent sections for individual datasets and experiments giving the details on specific licenses and terms of use.

Guidelines:

- The answer NA means that the paper does not use existing assets.
- The authors should cite the original paper that produced the code package or dataset.
- The authors should state which version of the asset is used and, if possible, include a URL.
- The name of the license (e.g., CC-BY 4.0) should be included for each asset.
- For scraped data from a particular source (e.g., website), the copyright and terms of service of that source should be provided.
- If assets are released, the license, copyright information, and terms of use in the package should be provided. For popular datasets, `paperswithcode.com/datasets` has curated licenses for some datasets. Their licensing guide can help determine the license of a dataset.
- For existing datasets that are re-packaged, both the original license and the license of the derived asset (if it has changed) should be provided.
- If this information is not available online, the authors are encouraged to reach out to the asset's creators.

13. **New assets**

Question: Are new assets introduced in the paper well documented and is the documentation provided alongside the assets?

Answer: [Yes]

Justification: New XAI-TRIS datasets (Clark et al., 2024b) are introduced in the main text with the full generation and usage details available in the Appendix Section D. Data preprocessing and usage instructions for the real world examples COMPAS (ProPublica, 2016) and MIMIC-IV (Johnson et al., 2023b) are stated in their respective results with the full instructions given in the appendices. All generation and preprocessing assets are available with the code provided.

Guidelines:

- The answer NA means that the paper does not release new assets.
- Researchers should communicate the details of the dataset/code/model as part of their submissions via structured templates. This includes details about training, license, limitations, etc.
- The paper should discuss whether and how consent was obtained from people whose asset is used.
- At submission time, remember to anonymize your assets (if applicable). You can either create an anonymized URL or include an anonymized zip file.

14. **Crowdsourcing and research with human subjects**

Question: For crowdsourcing experiments and research with human subjects, does the paper include the full text of instructions given to participants and screenshots, if applicable, as well as details about compensation (if any)?

Answer: [NA]

Justification: The paper uses existing datasets COMPAS (ProPublica, 2016) and MIMIC-IV (Johnson et al., 2023b) which themselves make use of data human subjects. All details can be seen in the original source materials.

Guidelines:

- The answer NA means that the paper does not involve crowdsourcing nor research with human subjects.
- Including this information in the supplemental material is fine, but if the main contribution of the paper involves human subjects, then as much detail as possible should be included in the main paper.
- According to the NeurIPS Code of Ethics, workers involved in data collection, curation, or other labor should be paid at least the minimum wage in the country of the data collector.

15. **Institutional review board (IRB) approvals or equivalent for research with human subjects**

Question: Does the paper describe potential risks incurred by study participants, whether such risks were disclosed to the subjects, and whether Institutional Review Board (IRB) approvals (or an equivalent approval/review based on the requirements of your country or institution) were obtained?

Answer: [NA]

Justification: For the MIMIC-IV dataset, given the de-identified nature of the data, the Beth Israel Deaconess Medical Center's ethical committee waived the requirement for informed consent (Johnson et al., 2023b). For the COMPAS data, there are known risks and biases including racial biases (Angwin et al., 2016; Dressel & Farid, 2018; Tan et al., 2018), which we discuss in the main text including the risk of unfair treatment as a result. However, we do not provide new data or experiments of our own that involve human subjects, so we have answered NA to this and the above questions.

Guidelines:

- The answer NA means that the paper does not involve crowdsourcing nor research with human subjects.
- Depending on the country in which research is conducted, IRB approval (or equivalent) may be required for any human subjects research. If you obtained IRB approval, you should clearly state this in the paper.
- We recognize that the procedures for this may vary significantly between institutions and locations, and we expect authors to adhere to the NeurIPS Code of Ethics and the guidelines for their institution.
- For initial submissions, do not include any information that would break anonymity (if applicable), such as the institution conducting the review.

16. **Declaration of LLM usage**

Question: Does the paper describe the usage of LLMs if it is an important, original, or non-standard component of the core methods in this research? Note that if the LLM is used only for writing, editing, or formatting purposes and does not impact the core methodology, scientific rigorousness, or originality of the research, declaration is not required.

Answer: [Yes]

Justification: We have stated our usage of LLMs at the start of the Results section, and have ticked the relevant checkboxes on OpenReview to disclose this, including: Editing (e.g., grammar, spelling, word choice), Data processing/filtering, Visualizing results for submission, Facilitating or running experiments, Implementing standard methods.

Guidelines:

- The answer NA means that the core method development in this research does not involve LLMs as any important, original, or non-standard components.
- Please refer to our LLM policy (`https://neurips.cc/Conferences/2025/LLM`) for what should or should not be described.

