# OpenReview forum: "Correcting misinterpretations of additive models"
_NeurIPS.cc/2025/Conference — NeurIPS 2025 poster_

### Official Review · Reviewer_3mR9 · 2025-06-18

**Clarity:** 4
**Significance:** 3
**Originality:** 3
**Rating:** 5
**Confidence:** 4

**Summary:**

This work proposes a novel post-processing method for generalized additive models (GAMs) to account for data covariance and thereby lessen the influence of suppressor variables on explanations derived from the GAM. This is achieved by extending the idea of activation patterns by Haufe et al. from linear models to polynomial feature spaces (PatternQLR) and GAMs (PatternGAM). In PatternQLR, this work maps (invertibly) the data into a non-redundant quadratic feature space (similar in nature to a degree-2 polynomial kernel, only with redundant terms removed) then trains a linear classifier. Similarly, in PatternGAM, this work maps the data into a feature space determined by the component functions (both marginal and interaction components) of a GAM. In the experiments in this work, these component functions are neural additive models with either one or two-input subnetworks, for marginal and interaction components, resp. This work then trains a logistic regression model over the mapped features, computes activation pattern vectors, and thereby obtains pattern shape functions by inverting the model output and applying the activation pattern vector. These pattern shape functions incorporate information about the covariance in either quadratic or GAM feature spaces, and can thereby identify and ignore suppressor variables. Experiments, on both synthetic and real data, are provided to verify the algorithms' efficacy, along with a novel evaluation metric.

**Questions:**

1. In the real-world case studies, I was surprised by the use of an ensemble of 100 NAMs, where this was not mentioned before. Why was an ensemble used, and not a single NAM? Also, is it possible to represent an ensemble of 100 NAMs, each using the same features, as a single NAM via a weighted feature-wise combination of the individual NAMs in the ensemble?

2. I found it interesting that, in both real-world case studies, interaction terms did not meaningfully improve the predictive power of the models. I have several subquestions in this direction. Does the underlying NAM ensemble, before applying PatternGAM, achieve comparable generalization performance to the NAM with interactions, or does PatternGAM improve the generalization performance to the level of an interaction NAM? Have you tested this phenomenon on other datasets? If so, does it seem to generally hold that interaction terms aren't useful on so-called 'real-world' tabular data?

3. Could the use of LASSO regularization in the logistic regression problem solved in creating pattern activation vectors help to consolidate predictive power into the smallest possible number of variables, and further eliminate the influence of sensitive features in the case studies? Or would this be somehow counterproductive?

4. Related to the previous question, how does this method compare in ignoring the influence of suppressor variables to sparse GAM methods, such as FastSparse? Intuitively, sparsity regularization during training would achieve a similar objective by forcing the optimization algorithm to consolidate predictive power into as few features/splits as possible.

5. In the definition of FNI-EMD, what motivated the choice to identify a ground-truth interaction as one where both marginal features are present? It seems like this over-represents the set of ground-truth interactions. Also, does FNI-EMD prefer subset-style explanations over complete ones, or does it merely refrain from additionally penalizing subsets?

Bonus question out of curiosity: On the synthetic vision problem, can the results obtained by this method be compared to an interpretable case-based reasoning neural network, such as ProtoPNet?

**Ethical Concerns:**

["NO or VERY MINOR ethics concerns only"]

**Final Justification:**

My discussion with the authors revealed the following issues, all of which were resolved:
1. It was not clear whether PatternGAM was a predictive method or an explanation method.
2. There was not theoretical justification in the paper itself for why the proposed method indeed fixes the influence of suppressor variables.
3. The figures in the paper were not incorrect, but led me to make incorrect conclusions about the paper.

The authors have proposed good changes and additions to the camera ready version of the paper that
- will not substantively change the paper, but
- will make the message much clearer and the contribution easier to understand.

With that in mind, I am keeping my recommendation to accept the paper, and increasing my confidence in acceptance.

**Limitations:**

yes

**Quality:**

3

**Strengths And Weaknesses:**

## Strengths

1. The problem statement is very well-motivated, and the proposed solution is demonstrated to be, via comparison to related work, a novel and interesting way to address the detrimental effect of suppressor variables on GAM interpretability. The allure of a kernel-based method to solve this problem was misleading, as the authors identified, because it was not easily invertible. The proposed solution seems just as powerful (in the quadratic case, it only eliminates redundant/symmetric terms) and is easily invertible to recover precise pattern shape functions.

2. I appreciated that this paper did not claim to solve all problems with GAM shape function-based explanations, such as confounder variables and data artifacts like default missing values. The focus of the paper on suppressor variables helped to justify the methodological choices and led to very interesting experiments which demonstrate clear practical utility and significance of the proposed method.

3. The synthetic experiments, along with the proposed false-negative invariance distance metric, are convincing and interesting in their own right. Figure 1 clearly demonstrates that the pattern-based method successfully ignores the incorrect influence of many irrelevant features to the prediction task, and Figure 2 gives a striking visual which clearly shows the very intuitive interaction patterns used by PatternGAM to solve the XOR problem with tetrominos. These interaction patterns seem to be a) clearly the correct way for a model to reason about the problem, b) unobtainable by previous GAM methods, and c) unexplainable by a non-interpretable vision model.

4. The COMPAS and MIMIC-IV examples in the text were convincing and demonstrated that this approach can give novel insights about real data which were not possible via previous methods.

## Weaknesses

### Significant Weaknesses

1. I felt that a coarse theoretical argument detailing why pattern-based methods are able to ignore suppressor variables would have improved the paper. I recognize that suppressor variables have yet to be clearly defined by the literature, and this is clearly identified by the authors. However, I feel as though a more clear working definition, backed up by references to theory on linear models from prior work, could have helped to ground the intuitive arguments provided by this work for the proposed methods. The example provided in the background is helpful, but lacks a more general claim to justify the activation pattern approach as fixing the suppressor variable problem. As it stands, the reader must trust that incorporating data covariance into the model solves the problem (as it does in the linear case), which is intuitive to me but not rigorous.

2. Table 1 only weakly supports the claim that PatternGAM significantly outperforms past methods on the FNI-EMD metric. Both PatternAttribution and Int. Grads. achieve comparable performance to PatternGAM on most scenarios, and both of them beat PatternGAM on every XOR scenario. I respect that the authors did not cherry-pick results, and I do not think that this invalidates the novelty or significance of the paper -- however, I do think that this weakens the claim that PatternGAM is uniquely suited to generating faithful subset explanations and avoiding false positives compared to other state of the art methods.

### Minor Weaknesses

1. Without having read the appendix to learn more about the data generation process, the orange boxes in Figure 1 were very hard to understand. Also, incorrect explanations happen outside of the orange boxes in Figure 1, which I found to be confusing if the purpose of this method is to eliminate distractors (it's eliminating more than just distractors). After reading the appendix, the figure made a lot more sense.

2. The analysis of the COMPAS task in section 4.2 seemed to slightly overstate the proposed method's capabilities to ignore influence of sensitive features. This approach only applies if the sensitive features happen to be suppressor variables.

---

> ### Author Rebuttal · Authors · 2025-07-31
>
> **Significant Weakness 1:**
>
> We appreciate the reviewer’s thoughtful suggestion. We agree that a more explicit theoretical argument and working definition for suppressor variables would strengthen the paper.
>
> In the revised version, we will:
>
> 1. Provide a clear working definition of suppressor variables in the context of additive models, grounded in prior literature (e.g., Conger, 1974;  Haufe et al., 2014; Wilming et al., 2023). In our context, we define a suppressor as a variable that improves predictive performance by carrying information about other features, but has no statistical dependency with the target variable. This contrasts with informative variables, which are directly associated with the label.
>
> 2. Formally demonstrate that PatternGAM avoids attributing importance to such suppressor variables, based on the Statistical Association Property (SAP) introduced by Wilming et al., (2023). Specifically, we will show that the pattern value - which reduces to the covariance between the nonlinearly transformed feature and the target - is zero when no statistical dependency exists. This allows PatternGAM to disambiguate suppressors and noise from confounders and informative features.
>
> 3. Expand our intuitive explanation with a more general argument: when the NAM learns a linearly separable representation from potentially nonlinear input space, PatternGAM projects the learned decision boundary back through the data distribution.
>
> Conger, A. J. (1974). A Revised Definition for Suppressor Variables: a Guide To Their Identification and Interpretation. Educational and Psychological Measurement, 34(1), 35-46
>
> Haufe, Stefan, et al. "On the interpretation of weight vectors of linear models in multivariate neuroimaging." Neuroimage 87 (2014): 96-110.
>
> Wilming, Rick, et al. "Theoretical behavior of XAI methods in the presence of suppressor variables." International Conference on Machine Learning. PMLR, 2023.
>
>
> **Significant Weakness 2:**
>
>
> PatternGAM is designed to explain additive models such as NAMs or EBMs. In the XOR scenarios, for example, the NAM architecture itself exhibits limited predictive performance compared to the MLP (see Figure 5), reflecting a mismatch between model capacity and data complexity. SHAP, PatternAttribution, and Integrated Gradients are applied to the MLP that perform better on these tasks. Even with near perfect classification performance, methods like SHAP and Integrated Gradients have been shown analytically to assign importance to suppressor features (Wilming et al., 2023), and PatternAttribution has shown suboptimal performance in non-linear tasks where suppressors are present (Clark et al., 2024).
>
> Thus, while attribution methods on MLPs may appear to outperform PatternGAM when evaluated the FNI-EMD metric, this is in part a result of comparing explanations across fundamentally different model types. When restricted to additive model architectures, PatternGAM offers significant advantages in not attributing false positive importance to suppressors, as we demonstrate in both synthetic and real-world settings.
>
>
> **Minor weakness 1:**
>
> Thank you for pointing this out. We agree that the current version of Figure 1 may be unclear without the context provided in the appendix, particularly regarding the meaning of the orange boxes. To clarify:
>
> 1. The orange boxes indicate distractor regions - i.e., suppressor features that carry no statistical association with the target but influence the model’s decision through correlations with informative features.
>
> 2. We will update the figure caption and lines 230–234 to explicitly define the orange regions and distinguish them from noise or overlapping regions. Additionally, we will clarify that PatternGAM suppresses not only distractors (suppressors), but also any features with no statistical association to the target.
>
> **Minor weakness 2**
>
> Thank you for raising this point. Our intent was not to overstate this capability but to highlight an interesting outcome in the COMPAS task: PatternGAM neutralises the attribution to race and sex, suggesting that, in this model, these features behave more like suppressors than confounders or true predictors of recidivism. We will revise the text in Section 4.2 to more clearly distinguish between suppressors and irrelevant features, which PatternGAM is explicitly designed to identify and suppress; versus confounders, which may still receive attribution if they are statistically associated with the label.
>
> **Question 1:**
>
> We can see in Figures 3 and 4 that the many lighter shaded lines are individual models from the ensemble, showing the variation and stochastic nature of the NAM optimisation process. The emboldened line can be seen as the ‘mean function’, or the reviewer’s idea of a weighted feature-wise combination. Taking an ensemble of many NAMs allows us to capture this average effect and account for such variation.
>
> **Question 2**
>
> Thank you for raising this interesting observation. First, we would like to clarify that PatternGAM is a post-hoc interpretability method, not a model architecture. It does not affect model performance - it interprets the predictions of an already trained additive model (NAM or EBM), including when interaction terms are present. We will revise the terminology in the final version to better reflect this distinction.
>
> In both COMPAS and MIMIC-IV, we observed that adding pairwise interaction terms yielded only minor improvements in AUROC (e.g., 0.821 without interactions and  0.824 with interactions for the MIMIC-IV setting). This suggests that the majority of predictive signal in these tasks is already captured by univariate nonlinear effects, and the added value of interactions is limited due to redundancy or diminishing returns. For instance, in MIMIC-IV, many features (e.g., age, prior admissions, lab values) are already strong individual predictors.
>
> **Question 3**
>
> As discussed in Haufe et al. (2014), sparsity-inducing regularisation does not eliminate suppressors; in fact, suppressors can persist in even the sparsest optimal models if they help cancel noise in correlated predictors. PatternGAM’s goal is not to remove suppressors from the model, but to avoid assigning significant importance in explanatory importance. While LASSO could produce more compact explanations, it would not distinguish suppressors from true predictors, and may bias the pattern attribution undesirably.
>
>
> **Question 4**
>
> While sparse GAM methods such as FastSparse aim to improve interpretability by concentrating predictive power into a minimal subset of features, they do not eliminate suppressor variables. As shown in Haufe et al. (2014), suppressors can remain essential for optimal predictive performance and may be retained even under heavy sparsity constraints. PatternGAM addresses this and will not attribute importance to suppressors.
>
> In that sense, PatternGAM complements sparse GAMs: it can be applied to any trained additive model, including sparsified ones, to produce corrected attributions that distinguish true signal from suppressors.
>
>
> **Question 5:**
>
> Regarding Equation (7): the matrices $\gamma_{uv}$​ and $C_{uv}$​ do not refer to interactions between pixels $u$ and $v$. Instead, they define the mass flow and cost of transporting explanation values between individual pixels in the FNI-EMD metric. In this formulation, the ground truth consists of individual important pixels, and the transport cost is set to zero between any pair of ground-truth pixels, regardless of spatial distance. This allows the metric to ignore false negatives, as intended.
>
> Regarding the ground cost matrix C′ for interactions: you're correct that the current draft does not specify this in full detail, which we will update. For interaction terms (e.g., $Z_{ij}$​), the ground distance is defined as the sum of Euclidean distances between both constituent pixels.
>
> **Also, does FNI-EMD prefer subset-style explanations over complete ones, or does it merely refrain from additionally penalizing subsets?**
>
> FNI-EMD does not prefer subset-style explanations over complete ones - it simply refrains from penalising explanations that are subsets of the ground truth. FNI-EMD transforms any explanation into the closest subset of the ground-truth support (effectively projecting the explanation onto the ground-truth region), and then computes the transport cost needed to move the attribution mass to that subset. This projection incurs zero cost, so any explanation that is a subset of the ground truth - whether partial or complete - receives the same (perfect) FNI-EMD score.
>
> **Bonus question out of curiosity: On the synthetic vision problem, can the results obtained by this method be compared to an interpretable case-based reasoning neural network, such as ProtoPNet?**
>
> Thank you for the interesting question. We believe that PatternGAM and ProtoPNet serve fundamentally different explanatory goals. PatternGAM provides feature-level attributions, identifying which parts of the input contribute to the model’s decision, with respect to the underlying data distribution. In contrast, ProtoPNet offers a case-based reasoning approach, explaining predictions by referencing similar training examples or learned prototypes.

---

> > ### Comment · Reviewer_3mR9 · 2025-08-04
> >
> > Thank you for the thorough response! All of my concerns are satisfied by the discussion and by the proposed changes. I remain convinced that this is a strong contribution, and I will maintain my score as such.
> >
> > Thank you for clarifying my misunderstanding of the FNI-EMD metric -- this discussion was helpful for me to see the utility of this metric.
> >
> > I think the paper will benefit from a more rigorous treatment of suppressor variables, and a clearer distinction in the discussion of experimental results regarding the difference between suppressor variables, irrelevant features, and confounders.
> >
> > One remaining point which I remain unsure of is whether PatternGAM is a post-hoc explanation method, or a post-processing method for GAMs to eliminate the influence of suppressor variables. I have a couple of follow-up questions out of curiosity, none of which will affect my score.
> >
> > 1. If I used the output of PatternGAM as my model, would it always make the same predictions as the base model?
> > 2. If not, how close are the predictions? Is there a guarantee that the PatternGAM "interpretation" of the base model is close to the base model?
> > 3. Is it possible to include some form of data-covariance aware regularization in the training process of GAMs so that they automatically ignore suppressor variables?

---

> > > ### Author Response · Authors · 2025-08-05
> > >
> > > Thank you for the confidence in our submission and the follow up points for discussion.
> > >
> > > **One remaining point which I remain unsure of is whether PatternGAM is a post-hoc explanation method, or a post-processing method for GAMs to eliminate the influence of suppressor variables. I have a couple of follow-up questions out of curiosity, none of which will affect my score.**
> > >
> > > **1. If I used the output of PatternGAM as my model, would it always make the same predictions as the base model? 2. If not, how close are the predictions? Is there a guarantee that the PatternGAM "interpretation" of the base model is close to the base model?**
> > >
> > > PatternGAM is not designed to approximate the model’s predictions - its role is interpretive, not predictive. However, when we construct the surrogate model using the post-hoc logistic regression (e.g., using $\mathbf{w}_\mathbf{Z}^\top \mathbf{Z}$), this surrogate will approximate the base model, since the NAM architecture produces a linearly separable representation. In that case, fidelity can be measured (e.g., via AUROC between surrogate and base model outputs), but it is not guaranteed to be perfect unless the model is exactly linear in the representation. If this surrogate model is not perfect, there could be some changes in predictions compared to the base model.
> > >
> > > The PatternGAM projection, on the other hand, is not predictive - it highlights which input features are statistically associated with the model's decision, not how the model arrives at the decision. We should note, and we will make this clearer in the revised manuscript, that we use the absolute-valued activation pattern
> > >
> > > $ \mathbf{a} = | \Sigma_\mathbf{Z} \mathbf{w}_\mathbf{Z} | $
> > >
> > > in the calculations shown on lines 180-184, such that we do not change the direction of the model's prediction as per $w_\mathbf{Z}$.
> > >
> > > **3. Is it possible to include some form of data-covariance aware regularization in the training process of GAMs so that they automatically ignore suppressor variables?**
> > >
> > > This is an interesting direction, but challenging. Suppressor variables help to improve prediction by cancelling noise in correlated predictors, even though they have no statistical association with the target. So from a purely predictive standpoint (e.g., minimising loss), they are often retained.
> > >
> > > You could imagine introducing data-covariance-aware regularisation that penalises reliance on components uncorrelated with the target, but doing so risks reducing accuracy and would bias the model away from the Bayes-optimal solution. The core insight of PatternGAM is that you don’t need to change the model to get better explanations - instead, you can correct the attributions post-hoc while keeping the predictive performance intact.
> > >
> > > The direction that you are suggesting might be formalised as a type of orthogonality constraint or information bottleneck, but it would go beyond the current scope of PatternGAM. It’s an interesting idea for future work.

---

> > > > ### Comment · Reviewer_3mR9 · 2025-08-05
> > > >
> > > > Ah -- thank you for clarifying. I think my confusion stemmed partly from the presentation of the PatternGAM projection in the figures as essentially another GAM -- I assumed that the projection was predictive because it was represented in the same way as the original GAM. Thank you for engaging in this discussion, I feel like I understand the core idea much better.
> > > >
> > > > It makes sense that regularizing in this direction would hurt performance, and cancelling noise in correlated predictors may even be beneficial to the predictor overall. In that sense, correcting attributions post-hoc seems to be the right way to go about the problem, even if it was possible to introduce a "perfect" version of the regularization to avoid suppressors.

---

> > > > > ### Author Response · Authors · 2025-08-05
> > > > >
> > > > > Thank you for the quick response and engagement with our rebuttals - we're glad that our comments have helped clarify your questions.
> > > > >
> > > > > You're right that the way PatternGAM is visualised in the figures (i.e., through shape functions) can give the impression that it defines a new predictive model. The pattern values (introduced on line 183) that are used to re-weight the underlying GAM shape function can be interpreted as corrected feature importance scores - zero for suppressors and irrelevant variables, non-zero for informative features.
> > > > >
> > > > > The reason we chose to preserve the shape function visualisations in the PatternGAM plots was to align with standard interpretability conventions in the GAM/NAM community. However, we now see how this might imply that PatternGAM is itself a predictive model, rather than a transformation of the original model's interpretive structure.
> > > > >
> > > > > In the revised manuscript, we will clarify this point and make the explanatory nature of these visualisations more explicit.
> > > > > An additional presentation could simply show the scalar importance scores as a table or bar chart, especially for tabular data. We will incorporate this in the revised manuscript (likely in the appendix as accompanying figures to the shape function figures such as Figures 1, 3, and 4) to avoid any visual ambiguity.
> > > > >
> > > > > Thanks again for engaging - your feedback has been really valuable!

---

> > > > > > ### Comment · Reviewer_3mR9 · 2025-08-05
> > > > > >
> > > > > > I think that's a very reasonable clarification and the proposed extra appendix figure could be very helpful in disambiguating the explanatory purpose of PatternGAM from the way that I misread it as a new predictive model.
> > > > > >
> > > > > > I'm glad that my feedback was helpful! Thank you for answering so many questions orthogonal to the reviewing score.

---

> > > > > > > ### Author Response · Authors · 2025-08-05
> > > > > > >
> > > > > > > We'll be happy to include the extra appendix figure.
> > > > > > >
> > > > > > > We've really enjoyed the discussion, so if you have any further questions then feel free to let us know. All the best and thanks once more!

---

### Official Review · Reviewer_Rw6c · 2025-07-01

**Clarity:** 3
**Significance:** 2
**Originality:** 2
**Rating:** 4
**Confidence:** 3

**Summary:**

The paper extends the “activation-pattern” correction developed for linear models to the broader family of additive models.  It introduces a covariance-aware explanation framework that yields two new attribution methods—PatternGAM (for generalised / neural additive models with pairwise interactions) and PatternQLR (for quadratic logistic regression)—and proposes a variant of Earth-Mover’s-Distance (FNI-EMD) that is invariant to false negatives.  Experiments on the XAI-TRIS synthetic benchmark, COMPAS recidivism scores, and MIMIC-IV mortality prediction show that PatternGAM/PatternQLR reduce spurious attributions to suppressor variables and sometimes nullify sensitive features while preserving accuracy.

**Questions:**

Please refer to Weaknesses.

**Ethical Concerns:**

["NO or VERY MINOR ethics concerns only"]

**Final Justification:**

The authors’ response has addressed my concerns, and I appreciate the clarifications provided. I have therefore increased my score accordingly.

**Limitations:**

yes

**Quality:**

2

**Strengths And Weaknesses:**

Strengths
1. Sound formulation of a covariance-aware correction for additive models; clear empirical protocol; publicly released code & data.

2. Paper is generally well written, with thorough appendix and checklist.

3. Addresses the practical problem of false-positive attributions in “interpretable” additive models; may inform fairness audits where NAMs are popular.

4. Adapts linear activation-pattern idea to a new model family; introduces FNI-EMD metric.

Weaknesses

1. The operation of premultiplying the weights by the covariance matrix requires a formal proof to guarantee its validity.

2. The computational complexity increases explosively with dimensionality, rendering the method hard to use in practice; the paper should explore approximate strategies that can alleviate this issue.

3. The empirical evaluation omits comparisons with several mainstream explanation techniques—such as SHAP, LIME, and Integrated Gradients—whose inclusion is necessary for a fair assessment.

4. Purely additive model structures are uncommon in real-world applications, which inherently restricts the practical reach of the proposed approach.

5. The method cannot remove dependencies stemming from nonlinear correlations; although the authors acknowledge this limitation, it remains important to analyse how influential such dependencies become in more complex settings.

---

> ### Author Rebuttal · Authors · 2025-07-31
>
> **Weaknesses**
>
> **1. The operation of premultiplying the weights by the covariance matrix requires a formal proof to guarantee its validity.**
>
> This is straightforward to show. In the revised paper, we will explicitly define the Statistical Association Property (SAP), as formalised by Wilming et al. (2023). This property ensures that features with zero statistical dependency on the label (even nonlinearly) receive zero attribution. In the revised paper, we will provide a formal proof that the covariance adjusted shape functions and weights have the SAP property. We will show this for main effect (non-interaction) features, however extending the SAP to interaction features is non-trivial and will be an interesting direction for future work.
>
> Wilming, Rick, et al. "Theoretical behavior of XAI methods in the presence of suppressor variables." International Conference on Machine Learning. PMLR, 2023.
>
>  **2. The computational complexity increases explosively with dimensionality, rendering the method hard to use in practice; the paper should explore approximate strategies that can alleviate this issue.**
>
> We appreciate the reviewer’s concern regarding scalability. While the presentation of PatternGAM using covariance-based correction may give the impression of high computational complexity, the method is efficient in practice. Specifically, PatternGAM reduces to computing the covariance between each (individual) nonlinearly transformed feature (or interaction term) and the target, which can be done without explicitly forming the full covariance matrix or computing matrix-vector products. We will include a formal derivation of this in the revised version.
>
> **3. The empirical evaluation omits comparisons with several mainstream explanation techniques—such as SHAP, LIME, and Integrated Gradients—whose inclusion is necessary for a fair assessment.**
>
> We thank the reviewer for the feedback. However, we would like to clarify that both SHAP and Integrated Gradients are included in our empirical evaluation. These methods are reported in the final two columns of Table 1 (as well as Tables 2 and 3 in the appendix), and correspond to columns 7 and 8 in Figure 6.
>
> While we did not include LIME in our evaluation, this was a deliberate choice. Prior work (e.g., Wilming et al., 2022, 2023; Clark et al., 2024) has highlighted that LIME often fails to recover informative features in synthetic benchmarks with structured ground truth, such as ours. We therefore prioritised attribution methods with stronger theoretical foundations or empirical performance in comparable settings. We will clarify this decision and cite supporting work in the revised manuscript.
>
> Clark, Benedict, Rick Wilming, and Stefan Haufe. "XAI-TRIS: non-linear image benchmarks to quantify false positive post-hoc attribution of feature importance." Machine Learning 113.9 (2024): 6871-6910.
>
> **4. Purely additive model structures are uncommon in real-world applications, which inherently restricts the practical reach of the proposed approach.**
>
> We appreciate the reviewer’s point that purely additive model structures are less common in unconstrained real-world modeling. However, additive models are  highly relevant and widely used in recent practice ,particularly in domains that require interpretable or auditable decision-making. For example, Caruana et al. (2015) showed that an intelligible GAM with pairwise interactions could match black-box model accuracy in predicting pneumonia risk while surfacing and correcting clinically implausible patterns. Ravindra et al., (2019) studied multiple applications of GAMs to link air pollution and climatic variability with adverse health outcomes. Yang et al. (2025) recently showed that GAMs can be effective for predicting energy savings based on measurement and verification of efficiency measures in commercial buildings, which is a key requirement of sustainability goals. In such contexts, avoiding false positive attributions to suppressor variables is essential, making PatternGAM’s contribution directly relevant to real-world deployment of interpretable models.
>
> In our paper, we demonstrate the applicability of PatternGAM on two real-world datasets: COMPAS, a widely used fairness benchmark involving criminal recidivism prediction, and MIMIC-IV, a large-scale clinical dataset representative of real-world ICU decision-making. These reflect genuine use cases where interpretability is critical, and where additive models like NAMs and EBMs are increasingly adopted as presumed "glassbox" alternatives to black-box predictors.
> in the Discussion section of the revised manuscript (around lines 323-342), we will make it more explicit that there has been a recent increased interest in tabular data and tabular foundation models, and link this to existing writing on the popularity of using additive models for tabular data.
>
> Caruana, Rich, et al. "Intelligible models for healthcare: Predicting pneumonia risk and hospital 30-day readmission." Proceedings of the 21th ACM SIGKDD international conference on knowledge discovery and data mining. 2015.
>
> Ravindra, Khaiwal, et al. "Generalized additive models: Building evidence of air pollution, climate change and human health." Environment international 132 (2019): 104987.
>
> Yang, Jian, et al. "Climate adaptive energy efficiency modeling using a generalized additive approach to optimize building performance across Chinese climate zones." Scientific Reports 15.1 (2025): 20088.
>
>
> **5. The method cannot remove dependencies stemming from nonlinear correlations; although the authors acknowledge this limitation, it remains important to analyse how influential such dependencies become in more complex settings.**
>
> We thank the reviewer for raising this point. However, we would like to clarify that PatternGAM is explicitly designed to detect and attribute non-linear dependencies between features and the target, as long as these are captured within the model structure -such as univariate nonlinearities or explicit interaction terms. This aligns with the assumptions of the underlying GAM or NAM model: if the model captures a given nonlinear dependency, PatternGAM will reflect it in the attribution.
> The method is not restricted to linear correlations. In fact, in our evaluations (e.g., on the XAI-TRIS datasets, which include structured noise and nonlinear distractors), PatternGAM performs robustly even in the presence of nonlinear  correlations, outperforming other methods in identifying true feature relevance. We also include the non-linear dependence measures Mutual Information and the Hilbert-Schmidt Independence Criterion (HSIC) in Table 4 in the appendix to show that dependencies captured are not merely linear.
> To clarify, we do not claim in the paper that PatternGAM is limited to linear correlations, nor does the method filter them out - it highlights statistical associations as learned by the model, including nonlinear ones. If a GAM is not expressive enough to capture complex dependencies (e.g., beyond included pairwise or higher-order terms), then both model performance and explanation quality may degrade. In such cases, we recommend that explanations only be interpreted for well-performing models, where the learned representation is likely faithful.

---

> > ### Author Response · Authors · 2025-08-05
> >
> > Thank you again for your review and the feedback on our paper. We noticed that the mandatory acknowledgement of our rebuttal has not yet been submitted, and want to kindly follow up.
> > In our rebuttal, we clarified several points that we believe address the key weaknesses cited in the review. In particular, we would like to reiterate two of the key points from our rebuttal:
> >
> > 1. SHAP and Integrated Gradients were indeed included in our empirical comparison (see Table 1 and Appendix Tables 2-3 and Figure 6).
> >
> > 2. PatternGAM is designed to eliminate suppressor features (those only conditionally associated with the target) while correctly attributing importance to features with marginal association, linear or non-linear. We also include Mutual Information and HSIC scores in Table 4 to quantify these dependencies beyond simple linear correlation.
> >
> > We would be grateful if you could review our rebuttal and let us know if any concerns remain. If any part of our response is unclear or unsatisfactory, we’re happy to clarify further. Thanks again for your time and engagement.

---

> > ### Comment · Reviewer_Rw6c · 2025-08-05
> >
> > The authors’ response has addressed my concerns, and I appreciate the clarifications provided. I have therefore increased my score accordingly.

---

### Official Review · Reviewer_Xc1z · 2025-07-03

**Clarity:** 4
**Significance:** 3
**Originality:** 2
**Rating:** 4
**Confidence:** 2

**Summary:**

This paper identifies and addresses a critical shortcoming in how additive models—commonly perceived as inherently interpretable—are typically interpreted. It shows that such models, including generalized additive models (GAMs), neural additive models (NAMs), and explainable boosting machines (EBMs), can misattribute importance to suppressor variables, leading to misleading global explanations.

To address this, the authors introduce: 1. PatternQLR, which applies activation pattern correction to quadratic logistic regression; 2. PatternGAM, a generalization of the activation pattern framework that corrects global explanations in complex additive models via covariance-aware, post-hoc linearization of model outputs.

They further propose False-Negative Invariant Earth Mover’s Distance (FNI-EMD) as a metric better suited to assessing the quality of global explanations, especially in scenarios where explaining only a subset of ground truth features should not be penalized.

Experiments on synthetic (XAI-TRIS) and real-world datasets (COMPAS and MIMIC-IV) validate that PatternGAM outperforms both raw model weight inspection and post-hoc attribution methods (e.g., SHAP, Integrated Gradients), particularly in scenarios involving suppressors or correlated noise.

**Questions:**

1. The current formulation and evaluation of PatternGAM focus entirely on global explanations—i.e., producing a single explanation per class or model, rather than per input. Have you considered extending PatternGAM to the local explanation setting (e.g., similar in spirit to LIME or SHAP)? Specifically, could a covariance-aware linear projection be performed around a local neighborhood of an instance to produce a corrected explanation?

2. PatternGAM uses a post-hoc linear regression over shape function outputs to recover corrected attributions. While this is conceptually elegant and efficient, it may be sensitive to model complexity. How do you assess or control the fidelity of the linear approximation to the underlying model? Could this lead to under- or over-attribution in highly non-linear regions?

3. Your work extends naturally to models with pairwise interactions (e.g., EBM, NAM with pairwise terms). However, it is unclear whether the method generalizes beyond this. Could the covariance correction be generalized to additive models with three-way or higher-order interaction terms? If so, how would the linear regression stage be modified to accommodate this structure?

4. PatternGAM does not merely provide attributions—it offers corrected estimates that may be more robust to confounding. This opens up downstream opportunities. Could PatternGAM be used to guide feature selection (e.g., pruning suppressors), or to identify potential fairness issues in model behavior? Have you explored this in practice?

5. FNI-EMD is a compelling metric that addresses a real weakness in common evaluation metrics (e.g., IMA, EMD) by ignoring false negatives. However, its formulation and examples are tightly coupled to spatial data (e.g., 2D tetromino patterns). Could FNI-EMD be extended to structured tabular data, categorical features, or time series? Or is it specific to grid-based visual domains?

**Ethical Concerns:**

["NO or VERY MINOR ethics concerns only"]

**Limitations:**

Partially addressed. The authors note that their method is restricted to additive models and focuses on correcting statistical rather than causal misinterpretation. However, there is little discussion of: 1. Deployment risk: In real-world settings (e.g., COMPAS), even corrected attributions can be misunderstood by users. 2. Interpretability trade-offs: No analysis is provided on how users interpret PatternGAM explanations compared to other baselines. 3. Generality: The paper does not explore how to generalize PatternGAM to more complex architectures beyond NAMs and EBMs.

**Paper Formatting Concerns:**

1. Some heatmaps and shape function plots could be annotated more clearly (e.g., with bounding boxes for ground truth features).
2. Mathematical notation is generally clean, though parts of Section 3.1 may benefit from a brief intuitive paraphrase alongside the formal derivations.
3. The appendices are rich and well-documented; the data generation and classifier setup in XAI-TRIS are impressively thorough.

**Quality:**

3

**Strengths And Weaknesses:**

Strengths

The paper surfaces an underappreciated issue in additive model interpretation—spurious attribution due to suppressor variables—and offers a principled correction.

PatternGAM is built upon solid mathematical foundations, extending the activation pattern correction to non-linear models in a consistent and interpretable manner.

Across a range of scenarios, especially under correlated or confounded inputs, PatternGAM produces explanations that more accurately reflect known ground truth relevance.

The experimental results span synthetic, criminal justice, and medical domains, showing consistent advantages of the proposed method in both performance metrics and qualitative heatmaps.

FNI-EMD is a well-motivated addition to the XAI evaluation toolkit, avoiding false penalties for partial but correct explanations.

Weaknesses

The method is tailored to additive model families (GAMs, NAMs, EBMs) and cannot directly be applied to more expressive or end-to-end deep models (e.g., CNNs, transformers).

While PatternGAM adjusts explanation attribution, it doesn’t alter model training or prediction. The approach could miss interpretability issues that arise during optimization or from model biases.

PatternGAM is positioned as a global explanation tool, but no discussion is provided on whether or how it might be adapted to instance-level interpretability.

Like most attribution methods, PatternGAM is not causal. In high-stakes domains like recidivism or healthcare, users could potentially overinterpret covariance-aware corrections as causal insight.

---

> ### Author Rebuttal · Authors · 2025-07-31
>
> Thank you for the accurate and thoughtful summary of our work. We have had to remove a lot of explanation of our responses and have combined responses on repeated points/questions (for example, generality of PatternGAM, a local version of PatternGAM) due to rebuttal length constraints. Please feel free to ask for further clarification over the next week!
>
> **The method is tailored to additive model families (GAMs, NAMs, EBMs) and cannot directly be applied to more expressive or end-to-end deep models (e.g., CNNs, transformers).**
>
> We see PatternGAM as a first step toward bringing the soundness of the activation pattern framework to non-linear architectures. We chose (neural) additive models as a target because they are actively promoted as “intrinsically interpretable” models - yet, as we show, they are still vulnerable to suppressor-induced misinterpretations. PatternGAM addresses this by correcting how importance is attributed post-hoc, without requiring inverse mapping assumptions.
> Generalising PatternGAM to end-to-end deep models like CNNs and transformer architectures would require a principled way to extract structured, linearly separable representations suitable for pattern-based transformations - a challenge we see as promising future work.
>
> **While PatternGAM adjusts explanation attribution, it doesn’t alter model training or prediction. The approach could miss interpretability issues that arise during optimization or from model biases.**
>
> We agree that PatternGAM, by design, does not alter model training or predictions - it operates entirely post-hoc to analyse learned representations. While it's unclear which specific interpretability issues the reviewer refers to, we would like to emphasise that PatternGAM possesses the Statistical Association Property (SAP), as formalised by Wilming et al. (2023). This property ensures that features with zero statistical dependency on the label (even nonlinearly) receive zero attribution. In contrast, methods like LIME, SHAP, LRP, or Integrated Gradients can attribute importance to suppressor variables. This makes PatternGAM particularly well-suited for detecting issues like confounding or the "Clever Hans" effect (Lapuschkin et al., 2019) where a model exploits spurious correlations in the data.
>
> **Like most attribution methods, PatternGAM is not causal. In high-stakes domains like recidivism or healthcare, users could potentially overinterpret covariance-aware corrections as causal insight.**
>
> This is also true. However, again due to the statistical association property, PatternGAM is better suited as a screening tool to study causal associations than any existing feature attribution method known to us (e.g., LIME, SHAP, LRP, IG). While the dependencies uncovered by PatternGAM do not dissociate causal, anticausal, or confounded relationships, they provide a principled starting point for downstream causal analysis or auditing. We emphasise this distinction in the revised paper to help guide appropriate interpretation in high-stakes domains.
>
>
> **Questions:**
>
> **1. The current formulation and evaluation of PatternGAM focus entirely on global explanations—i.e., producing a single explanation per class or model, rather than per input. Have you considered extending PatternGAM to the local explanation setting (e.g., similar in spirit to LIME or SHAP)? Specifically, could a covariance-aware linear projection be performed around a local neighborhood of an instance to produce a corrected explanation?**
>
> Thank you for this very thoughtful suggestion. In fact, we are exploring exactly that idea in a separate publication, for which we already published a preprint. We believe that the complexities that arise in the local setting due to the various design choices with respect to defining local neighbourhoods and covariances and the different possible applications and experiments (as the resulting methods are not restricted to tabular data but applicable to image classification tasks, for example), warrant a separate manuscript. Nevertheless, we will be happy to disclose our preprint here after the review process is completed for the reviewer’s interest (provided this is possible).
>
> **2. PatternGAM uses a post-hoc linear regression over shape function outputs to recover corrected attributions. While this is conceptually elegant and efficient, it may be sensitive to model complexity. How do you assess or control the fidelity of the linear approximation to the underlying model? Could this lead to under- or over-attribution in highly non-linear regions?**
>
> PatternGAM assumes, consistent with the GAM and NAM architectures, that the model prediction is a linear combination of nonlinearly transformed (per-feature) shape function outputs. This structural constraint ensures that a post-hoc linear regression over these outputs is not an approximation in the usual sense, but rather reflects the actual composition of the model. For more complex, non-additive interactions that cannot be expressed through individual or pairwise shape functions, the additive assumption may become limiting. In such cases, model performance may be degraded. We advocate that explanations should only be applied to well-performing models, where the output representation is then linearly separable.
>
> **3. Your work extends naturally to models with pairwise interactions (e.g., EBM, NAM with pairwise terms). However, it is unclear whether the method generalizes beyond this. Could the covariance correction be generalized to additive models with three-way or higher-order interaction terms? If so, how would the linear regression stage be modified to accommodate this structure?**
>
> Extending PatternGAM to handle higher-order interactions (e.g., three-way or more) is conceptually straightforward. These higher-order terms can be added as explicit components in the GAM or NAM architecture - either as handcrafted terms or learned sub-networks - and treated as additional features in both the model and the post-hoc PatternGAM attribution step. Importantly, PatternGAM does not require explicit construction of a high-dimensional covariance matrix. The attribution reduces to computing the covariance between each shape function output (whether a main effect or interaction term) and the target.
>
> **4. PatternGAM does not merely provide attributions—it offers corrected estimates that may be more robust to confounding. This opens up downstream opportunities. Could PatternGAM be used to guide feature selection (e.g., pruning suppressors), or to identify potential fairness issues in model behavior? Have you explored this in practice?**
>
> We agree that PatternGAM provides corrected explanations by disambiguating suppressors from informative features, and we appreciate the suggestion to explore its use for downstream tasks like feature selection or fairness auditing.
> However, we would like to clarify that by “correction” we refer strictly to correcting explanations, not altering the model itself. This distinction is crucial: Bayes-optimal models may rely on suppressor variables to improve predictive performance (as discussed by Wilming et al., 2023), and such variables should not necessarily be removed from the model.
>
> **5. FNI-EMD is a compelling metric that addresses a real weakness in common evaluation metrics (e.g., IMA, EMD) by ignoring false negatives. However, its formulation and examples are tightly coupled to spatial data (e.g., 2D tetromino patterns). Could FNI-EMD be extended to structured tabular data, categorical features, or time series? Or is it specific to grid-based visual domains?**
>
> While our empirical focus was on 2D spatial domains (e.g., XAI-TRIS), the FNI-EMD metric is not inherently limited to grid-based visual data. The core requirement for FNI-EMD is the ability to define a ground cost (distance) between features, which allows it to be extended to other domains, such as temporal distance for time series, Mahalanobis distance in feature space for structured tabular data, or Hamming distance for categorical features. We will clarify this distinction in the discussion section of the revised text, and will also mention the alternative ground metrics stated above for their respective problem domains in Section 3.2.
>
> **2. Interpretability trade-offs: No analysis is provided on how users interpret PatternGAM explanations compared to other baselines.**
>
> Thank you for the comment. We agree that understanding how users interpret explanations is important. While a formal user study is out of scope for this paper, we view it as a valuable direction for follow-up work and are actively exploring it. The standard NAM explanations (via shape functions) serve as a natural baseline, and our results demonstrate how PatternGAM corrects key misattributions caused by suppressor variables. We acknowledge that additional baselines (e.g., gradient-based or heuristic methods) may further contextualise our results, so we will include representative comparisons to such baselines in the final manuscript.
>
> **Paper Formatting Concerns:**
>
> Thank you for the suggestions. We will adopt these idea in the revised manuscript. For example, Figure 2 will outline the ground truth tetromino subplots, which are the same respective positions as in Figure 1. We will expand Section 3.1 to improve clarity by providing brief intuitive explanations alongside the formal derivations - particularly in the PatternGAM paragraph.
>
> References:
>
> Clark, Benedict, Rick Wilming, and Stefan Haufe. "XAI-TRIS: non-linear image benchmarks to quantify false positive post-hoc attribution of feature importance." Machine Learning 113.9 (2024): 6871-6910.
>
> Haufe, Stefan, et al. "Position: XAI needs formal notions of explanation correctness." Interpretable AI: Past, Present and Future, 2024.
>
> Wilming, Rick, et al. "Theoretical behavior of XAI methods in the presence of suppressor variables." International Conference on Machine Learning. PMLR, 2023.

---

> > ### Author Response · Authors · 2025-08-05
> >
> > Thank you again for your thoughtful and constructive review. We wanted to briefly follow up, as the discussion phase is nearing its end and we haven’t yet seen a response to our rebuttal.
> >
> > In our rebuttal, we aimed to directly address your comments regarding:
> > - generality of PatternGAM beyond additive models (Weakness 1),
> > - missing interpretability issues arising during optimisation or from model biases (Weakness 2),
> > - PatternGAM and XAI methods relating to causal insights (Weakness 3),
> > - the extension of PatternGAM to local explanations (Q1),
> > - robustness in highly nonlinear regions (Q2),
> > - generalisation to higher-order interactions (Q3),
> > - downstream use in fairness and feature selection (Q4),
> > - applicability of FNI-EMD to non-spatial domains (Q5),
> > - and we also acknowledged your point about user-centered interpretability evaluation and noted this as a priority for future work (Limitation 2).
> >
> > If anything remains unclear or you’d like further clarification, we’ll be happy to elaborate. Thank you again for your time and engagement.

---

> > ### Comment · Reviewer_Xc1z · 2025-08-06
> >
> > Thank you for the detailed rebuttal and clarifications regarding your submission. I appreciate your thoughtful engagement with the points raised.
> >
> > 1. Your rebuttal reiterates the key contributions of PatternGAM and PatternQLR as principled extensions of the activation pattern concept to additive models. I acknowledge the novelty in applying covariance-aware explanations to NAMs and GAMs, and the promising empirical results on XAI-TRIS and real-world datasets like COMPAS and MIMIC-IV. However, I remain concerned that certain key concepts—particularly the covariance-based reweighting in the nonlinear setting—require clearer formalization and intuition in the main text. While the rebuttal offers helpful clarifications, this depth of explanation would ideally be part of the paper itself. For instance, a more pedagogical introduction to suppressor variables in additive models (perhaps with a toy example or schematic) would greatly enhance accessibility for readers unfamiliar with this specific pathology.
> >
> > 2. The empirical results on synthetic and real datasets are compelling, especially with the introduction of FNI-EMD. However, the rebuttal does not sufficiently address my concern about broader applicability. On one hand, the method's reliance on well-behaved shape functions (e.g., NAMs with certain architectural constraints) may limit its robustness in real-world, high-dimensional settings. On the other hand, it is not clear how PatternGAM would perform when interaction terms are noisy, mis-specified, or learned spuriously in practice. While the rebuttal touches on these points briefly, more systematic ablations or sensitivity analyses would strengthen your empirical claims.
> >
> > 3. I appreciate your explanation regarding the nullification of race and sex features in the COMPAS case. However, I remain cautious about the implications. The rebuttal claims that these features act as suppressors rather than confounders—but this distinction is subtle and consequential. While PatternGAM's ability to “nullify” certain variables is a useful tool, care must be taken in interpreting what this nullification implies about the fairness or informativeness of the feature. The current language risks overstating the fairness implications without a deeper causal or fairness-aware analysis. This is not a fundamental flaw, but I suggest tempering such claims in future revisions.
> >
> > Overall, I continue to view this paper as a valuable and timely contribution to interpretable ML and XAI, particularly in addressing the suppressor effect in additive models. Nonetheless, I believe further work is required to improve clarity, broaden the evaluation, and provide stronger guidance on interpreting real-world applications.
> >
> > After carefully reviewing your responses, I have decided to maintain my original score.

---

> > > ### Author Response · Authors · 2025-08-08
> > >
> > > Thank you for your detailed follow-up and for recognising the novelty, empirical strength, and relevance of our work. We appreciate your constructive feedback and agree that the points you raise will strengthen the paper.
> > >
> > > To address your three remaining concerns:
> > >
> > >  1) Formalisation and pedagogical clarity: In the revised paper, we will explicitly define the Statistical Association Property (SAP), as formalised by Wilming et al. (2023). This property ensures that features with zero statistical dependency on the label (even nonlinearly) receive zero attribution. In the revised paper, we will provide a formal proof that the covariance adjusted shape functions and weights have the SAP property. We will show this for main effect (non-interaction) features, however extending the SAP to interaction features is non-trivial and will be an interesting direction for future work. Specifically, this will be in Section 2, where we will also include your suggestions of introducing suppressor variables directly in the context of additive models, and we will include a toy example and schematic illustration.
> > >
> > > 2) Broader applicability / robustness:
> > >
> > > 2a. Robustness (High-dimensional settings) - we would like to maintain the position that one should only interpret a performant model and so PatternGAM should only be applied when the model has generalised the underlying data. The NAM framework that we use allows for flexibility in the underlying architecture used for each feature, as well as controlled regularisation and dropout etc. With this and the flexibility of the ExU activation functions, we have observed that NAMs are very suitable to learn ‘wiggly’ and appropriate shape functions for high dimensional and complex real world data. Given prior knowledge of the input data/problem setting, one can even use different subnetworks for each feature. Any neural network-based architecture can be used, as long as the constraints of additive models are satisfied. We see, however, that this is not necessary for highly performant NAMs in the synthetic XAI-TRIS setting, or the two real-world settings of COMPAS and MIMIC-IV.
> > > Also, PatternGAM itself does not require explicit computation of a full covariance matrix. Instead, it only computes covariances between each shape function output and the target, which scales linearly with the number of model terms (main effects and interactions). Therefore, with a performant underlying NAM suited to the problem domain, we can guarantee the performance of PatternGAM both in terms of correctness and in terms of efficient computation. We will add discussion points relating to what we have said above - specifically about NAM architectural decisions and generality/flexibility, and how one can tailor NAMs and PatternGAM to their real world problems.
> > >
> > > 2b. Noisy or mis-specified interaction terms:
> > >
> > > The interaction terms themselves are chosen by the FAST algorithm, which  returns candidate interaction pairs based on estimating a minimal interaction model only on the residuals of the interaction pair - and evaluates how much residual sum of squares drops so to maximise downstream model performance. With a high enough number of interaction pairs $N$ specified to return for later NAM modelling, this will include suppressors (whose denoising role in multivariate modelling will increase model performance when present), and even uninformative (noise) features at a certain $N$.
> > > If an interaction from this set is purely noise or spurious, its statistical association with the target will be zero, and PatternGAM will assign it negligible attribution - effectively nullifying its explanatory influence.
> > > Internally, we have informally tested many sizes of feature interaction sets up to the full quadratic pairwise interaction set ($N=4096$ for XAI-TRIS), landing on $N=128$ interactions as a good balance of efficient computation without degraded model and explanation performance. We will formally test this to show variations in classification and explanation performance to put in the appendix of the revised manuscript. For a more systematic ablation experiment (e.g., injecting further forms of controlled noise into interaction terms, past the existing XAI-TRIS construction), we would welcome additional ideas of what else specifically you would like to see tested here.
> > >
> > > 3. Fairness implications in COMPAS:
> > > We will revise the wording around this experiment to explicitly avoid overstating fairness claims, framing the nullification of race/sex variables as a finding about their potential suppressor-like role in the trained model, without inferring broader causal or fairness conclusions.
> > >
> > > Given these points, and your stated view that the paper is a valuable and timely contribution, we hope you might consider whether an updated score would more accurately reflect the work’s contribution with the planned changes to the camera-ready manuscript.
> > >
> > > Thank you again for the engagement in the rebuttal process.

---

### Official Review · Reviewer_HbQD · 2025-07-05

**Clarity:** 3
**Significance:** 3
**Originality:** 3
**Rating:** 5
**Confidence:** 3

**Summary:**

This paper introduces PatternGAM and PatternQLR, two new explanation methods for additive models that correct for misinterpretations caused by suppressor variables—features that are not causally linked to the target but appear important due to correlations. By extending the activation pattern framework (originally for linear models) to additive models, the authors enable more faithful feature attributions in settings where standard methods fail. The proposed techniques are evaluated on both synthetic (XAI-TRIS) and real-world datasets (COMPAS and MIMIC-IV), demonstrating improved robustness and interpretability over popular XAI methods.

**Questions:**

Please check the weakness outlined above.

1. Does PatternGAM generalize to models beyond additive structures (e.g., deep networks, GNNs)?

**Ethical Concerns:**

["NO or VERY MINOR ethics concerns only"]

**Final Justification:**

I have read the rebuttal. I have already given positive score to this paper.  I would like to keep my score same.

**Limitations:**

Yes

**Quality:**

3

**Strengths And Weaknesses:**

**Strength**

1. **Timely and Important Topic:**  Tackles a critical issue in XAI—false-positive attributions caused by suppressors, especially relevant in high-stakes domains like healthcare and criminal justice.

2. **Novelty**: The extension of activation patterns to additive models is well-motivated. PatternGAM and PatternQLR provide a principled, covariance-aware explanation mechanism.

3. **Thorough Evaluation:** Strong empirical results across synthetic and real-world datasets. The introduction of False-Negative Invariant Earth Mover’s Distance (FNI-EMD) is a meaningful addition to the XAI evaluation toolkit.

**Weakness**

1. **Computational Overhead:**  Computing full covariance-based activation patterns, especially with interaction terms, may not scale well to very high-dimensional or real-time applications.

 2. PatternGAM tends to produce "subset explanations"—highlighting only some relevant features. While this is justified, it could limit applicability in settings requiring complete ground-truth attribution.

---

> ### Author Rebuttal · Authors · 2025-07-31
>
> Thank you for the accurate and thoughtful summary. Please note that we deliberately avoid framing our method in terms of faithfulness, as this concept (as commonly used in XAI) can systematically assign importance to suppressor features — a concern raised in recent work (e.g., Wilming et al., 2023). This is also discussed by Haufe et al. (2024). Instead, we focus on producing explanations that better reflect statistical signal attribution, disentangling predictive utility from spurious correlations introduced by suppressors.
>
>
> **Weakness**
>
>  **1. Computational Overhead: Computing full covariance-based activation patterns, especially with interaction terms, may not scale well to very high-dimensional or real-time applications.**
>
> Thank you for raising this. While computational overhead is a valid concern for explanation methods, PatternGAM is designed to be lightweight in practice. PatternGAM reduces to computing the covariance between each scalar-valued feature (or interaction) output and the target, rather than requiring a full covariance matrix or matrix-vector product. This can be computed efficiently using standard statistics over the dataset. Moreover, unlike methods such as LIME or SHAP, PatternGAM does not involve training local surrogate models or computing Shapley value approximations. The method is purely post-hoc and closed-form.
>
> If needed, approximate strategies such as diagonal/shrinkage covariance estimation, mini-batch statistics, or limiting the number of interaction terms (as we do via the FAST interaction selection algorithm) offer further scalability. We will make this more explicit by adding the above points to the discussion section in the revised version and provide bounds on the underlying computational complexity.
>
> **2. PatternGAM tends to produce "subset explanations"—highlighting only some relevant features. While this is justified, it could limit applicability in settings requiring complete ground-truth attribution.**
>
> Thank you for raising this point. We agree that PatternGAM can produce sparse or subset-style explanations, where only a subset of the relevant features receives non-zero attribution. This may reflect property of additive models like NAMs, where the model may shut down some redundant informative features through the function $f_i$, i.e. a feature’s shape function $f_i(x_i)$ collapses to a constant. Then, an informative feature $x_i$ would become an uninformative transformed feature $z_i$. Here, PatternGAM would correctly assign it zero attribution, even if that feature was relevant in the original data distribution. In this sense, sparsity in the explanation may not indicate incompleteness, but rather that the model has learned to ignore some redundant or weakly informative variables.
>
> **Questions:**
>
> **Please check the weakness outlined above.**
>
> **1. Does PatternGAM generalize to models beyond additive structures (e.g., deep networks, GNNs)?**
>
> Thank you for the question. PatternGAM, in its current form, is designed specifically for additive models. We see this as a first step toward bringing the soundness of the activation pattern framework to non-linear architectures. We chose (neural) additive models as a target because they are actively promoted as “intrinsically interpretable” models - yet, as we show, they are still vulnerable to suppressor-induced misinterpretations. PatternGAM addresses this by correcting how importance is attributed post-hoc, without requiring inverse mapping assumptions.
> While each subnet in a NAM can be made arbitrarily deep, and thus has some architectural flexibility, generalising PatternGAM to  deep models or graph neural networks would require a principled way to extract structured, linearly separable representations suitable for pattern-based transformations - a challenge we see as promising future work.
>
>
> References:
>
> Wilming, Rick, et al. "Theoretical behavior of XAI methods in the presence of suppressor variables." International Conference on Machine Learning. PMLR, 2023.
>
> Haufe, Stefan, et al. "Position: XAI needs formal notions of explanation correctness." Interpretable AI: Past, Present and Future, 2024.

---

### Note · Authors · 2025-08-12

We would like to thank the reviewers for their constructive engagement. Across the reviews, there is consensus that the paper presents a novel, principled extension of the activation pattern framework to additive models (NAMs, GAMs, EBMs), addresses the important and underexplored problem of suppressor-induced misattribution, and achieves compelling empirical results on both synthetic (XAI-TRIS) and real datasets (COMPAS, MIMIC-IV).

The remaining points raised are about clarity, scope, and additional evaluation - not methodological correctness. Here are a few of the key points raised during the rebuttal and discussion period:

1. Formalisation and clarity - In the camera-ready version we will: (i) explicitly define the Statistical Association Property (SAP) in Section 2, (ii) provide the formal proof that our covariance-adjusted shape functions satisfy the SAP for main effects, and (iii) add a pedagogical subsection with a toy example and schematic introducing suppressor variables in additive models.

2. Broader applicability and robustness - We will expand discussion of NAM architectural flexibility for high-dimensional settings, emphasising that PatternGAM scales linearly with model terms and only requires per-term covariances. For noisy/mis-specified interactions, PatternGAM’s statistical association property nullifies spurious terms.

3. Fairness interpretation - We will revise the COMPAS discussion to avoid overstating fairness implications, emphasising that the nullification of race/sex is potentially indicative of a suppressor-like role in the trained model, but is not a causal claim. We will add clarification on how PatternGAM is better suited towards causal reasoning in interpreting additive models than existing techniques, due to the Statistical Association Property (SAP).

4. Misunderstandings clarified - PatternGAM is not restricted to linear dependencies; it detects any dependency the model captures (including nonlinearities and interactions). SHAP and Integrated Gradients are included in our evaluation; LIME was excluded for reasons documented in the rebuttal.

These additions and the other changes that we have promised to make are straightforward to resolve in the camera-ready version, and do not require changes to the core methodology, results, or the conclusions that we draw. We believe that the contribution’s novelty, soundness, and relevance justify a positive decision.


Thank you for your time.

---

### Decision · Program_Chairs · 2025-09-17

**Decision:**

Accept (poster)

**Comment:**

The paper develops model explanation methods intended to disentangle features with statistical associations, and in particular to show dependence of additive models on features while controlling for "suppressor variables" (a type of / similar to confounder variables, depending on ones definition of confounding)

It builds off an approach called activation patterns, this pre-multiplies the vector of regression coefficients with the covariance matrix of predictor variables. The main contribution appears to be extending this approach to additive models in two ways, one by including interaction terms, and one by using GAM (generalized additive model) basis functions.

Reviewers were in agreement that the paper addressed an important problem and provided interesting empirical demonstrations. The discussion surfaced many questions which authors were able to answer sufficiently. Authors have committed to changes and additions that will clarify and strengthen the paper. These are all important, including more clearly defining suppressor variables (perhaps by using a formal causal framework like structural causal models)

I might add a suggestion for clarity: notation should clearly distinguish between (true) data generation models, predictive models, and explanation models. For example, a simple convention that could be helpful is to notice the difference between explaining $Y = f(X) +$ noise and explaining $\hat Y = \hat f(X)$.